# Optimize Planning Heuristics to Rank, not to Estimate Cost-to-Goal

**Leah Chrestien**
Czech Technical University in Prague
`leah.chrestien@aic.fel.cvut.cz`

**Tomáš Pevný**
Czech Technical University in Prague,
and Gen Digital, Inc.
`pevnytom@fel.cvut.cz`

**Stefan Edelkamp**
Czech Technical University in Prague
`edelkste@fel.cvut.cz`

**Antonín Komenda**
Czech Technical University in Prague
`antonin.komenda@fel.cvut.cz`

## Abstract

In imitation learning for planning, parameters of heuristic functions are optimized against a set of solved problem instances. This work revisits the necessary and sufficient conditions of strictly optimally efficient heuristics for forward search algorithms, mainly A* and greedy best-first search, which expand only states on the returned optimal path. It then proposes a family of loss functions based on ranking tailored for a given variant of the forward search algorithm. Furthermore, from a learning theory point of view, it discusses why optimizing cost-to-goal $h^*$ is unnecessarily difficult. The experimental comparison on a diverse set of problems unequivocally supports the derived theory.

## 1 Introduction

Automated planning finds a sequence of actions that will reach a goal in a model of the environment provided by the user. It is considered to be one of the core problems in Artificial Intelligence and it is behind some of its successful applications [39, 27, 46]. Early analysis of planning tasks [31] indicated that optimizing the heuristic function steering the search for a given problem domain towards the goal can dramatically improve the performance of the search. Automated optimization of the heuristic function therefore becomes a central request in improving the performance of planners [6, 13].

Learning in planning means optimizing heuristic functions from plans of already solved problems and their instances. This definition includes the selection of proper heuristics in a set of pattern databases [17, 22, 33, 12], a selection of a planner from a portfolio [25], learning planning operators from instances [32, 56], learning for macro-operators and entanglements [9, 29], and learning heuristic functions by general function approximators (e.g. neural networks) [45, 19, 16, 5].

The majority of research [45, 53, 19, 16] optimizes the heuristic function to estimate the cost-to-goal, as it is well known that it is an optimal heuristic for many best-first heuristic search algorithms including A* or IDA* [21, 28]. These works optimize the heuristic function to solve *regression problems*. But even a true cost-to-goal $h^*$ does not guarantee that the *forward search* will find the optimal solution while expanding the minimal number of states. This paper defines a stricter version of *optimally efficiency* [10, 24] as follows. *Forward search* (or its heuristic) is called *strictly optimally efficient* iff it expands *only* states on one optimal solution path returned by the search, i.e., if this optimal solution path has $l + 1$ states, the search will expand only $l$ states.

This work focuses exclusively on a *forward search* with merit functions. It presents theoretical arguments as to why the order (rank) of states provided by the heuristic function is more important

than the precise estimate of the cost-to-goal and it formalizes the necessary and sufficient condition for strictly optimally efficient search. It also argues why learning to rank is a simpler problem than regression estimating the cost-to-goal from a statistical learning theory point of view. The main motivation of the paper is similar to that in [38], but the derived theory and implementation are much simpler. The sufficiency of optimizing the rank has been already recognized in [59] for a beam-search and in [18] for greedy best-first search (GBFS), though, unlike this work, it cannot guarantee strict optimal efficiency.

We emphasize that this work neither deals with nor questions the existence of a strictly optimally efficient heuristic function with the forward search. It is assumed that the space of possible heuristic functions is sufficiently large, such that there exist one that is sufficiently good. The theoretical results are supported by experimental comparisons to the prior art on eight domains.

The contributions of this paper are as follows.

1. We state necessary and sufficient conditions for strictly optimally efficient heuristic functions for forward search algorithms in deterministic planning.

2. We show that when optimizing a heuristic function, it is better to solve the ranking problem.

3. We argue, why learning to rank is easier than regression used in learning the cost-to-goal.

4. We instantiate it for A* and GBFS searches leading to two slightly different loss functions.

5. We experimentally demonstrate on eight problems (three grid and five PDDL) that optimizing rank is *always* better.

## 2 Preliminaries

A search problem instance is defined by a directed weighted graph $\Gamma = \langle \mathcal{S}, \mathcal{E}, w \rangle$, a distinct node $s_0 \in \mathcal{S}$ and a distinct set of nodes $\mathcal{S}^* \subseteq \mathcal{S}$. The nodes $\mathcal{S}$ denote all possible states $s \in \mathcal{S}$ of the underlying transition system representing the graph. The set of edges $\mathcal{E}$ contains all possible transitions $e \in \mathcal{E}$ between the states in the form $e = (s, s')$. $s_0 \in \mathcal{S}$ is the initial state of the problem instance and $\mathcal{S}^* \subseteq \mathcal{S}$ is a set of allowed goal states. Problem instance graph weights (alias action costs) are mappings $w : \mathcal{E} \to \mathbb{R}^{\geq 0}$.

A path (alias a plan) $\pi = (e_1, e_2, \ldots, e_l)$, $e_i = (s_{i-1}, s_i) \in \mathcal{E}$, of length $l$ solves a task $\Gamma$ with $s_0$ and $\mathcal{S}^*$ iff $\pi = ((s_0, s_1), (s_1, s_2), \ldots, (s_{l-1}, s_l))$ and $s_l \in \mathcal{S}^*$. An optimal path $\pi^*$ is defined as a path minimizing the cost of a problem instance $\Gamma, s_0, \mathcal{S}^*$, its value as $f^* = w(\pi^*) = \sum_{i=1}^{l} w(e_i)$. When cost function $w(e) = 1$ optimal path corresponds to the shortest one. $\pi_{:i} = ((s_0, s_1), (s_1, s_2), \ldots, (s_{i-1}, s_i))$ denotes subplan of a plan $\pi$ containing its first $i$ actions. $\mathcal{S}^\pi = \{s_i\}_{i=0}^{l}$ denotes the set of states from the path $\pi = ((s_0, s_1), (s_1, s_2), \ldots, (s_{l-1}, s_l))$, and correspondingly $\mathcal{S}^{\pi:i}$ denotes the set of states in a subplan $\pi_{:i}$.

### 2.1 Forward search algorithm

To align notation, we briefly review the forward search algorithm, of which A* and GBFS are special cases. As we apply full duplicate detection in all of our algorithms, there is a bijection of search tree nodes to planning states. Forward Search for consistent heuristics, where $h(s) - h(s') \leq w(s, s')$ for all edges $(s, s')$ in the $w$-weighted state space graph, mimics the working of Dijkstra's shortest-path algorithm [11]. It maintains two sets: the first called *Open list*, $\mathcal{O}$, contains generated but not expanded nodes; the second called *Closed list*, $\mathcal{C}$, contains already expanded nodes. Parameters $\alpha$ and $\beta$ of a merit function $f(s) = \alpha g(s) + \beta h(s)$ used to sort states in Open list allow to conveniently describe different variants of the algorithm as described in detail shortly below. The forward search works as follows.

1. Initialise Open list as $\mathcal{O}_0 = \{s_0\}$.

2. Set $g(s_0) = 0$

3. Initiate the Closed list to empty, i.e. $\mathcal{C}_0 = \emptyset$.

4. For $i \in 1, \ldots$ until $\mathcal{O}_i = \emptyset$

   (a) Select the state $s_i = \arg\min_{s \in \mathcal{O}_{i-1}} \alpha g(s) + \beta h(s)$

(b) Remove $s_i$ from $\mathcal{O}_{i-1}$, $\mathcal{O}_i = \mathcal{O}_{i-1} \setminus \{s_i\}$

(c) If $s_i \in \mathcal{S}^*$, i.e. it is a goal state, go to 5.

(d) Insert the state $s_i$ to $\mathcal{C}_{i-1}$, $\mathcal{C}_i = \mathcal{C}_{i-1} \cup \{s_i\}$

(e) Expand the state $s_i$ into states $s'$ for which hold $(s_i, s') \in \mathcal{E}$ and for each

    i. set $g(s') = g(s_i) + w(s_i, s')$

    ii. if $s'$ is in the Closed list as $s_c$ and $g(s') < g(s_c)$ then $s_c$ is reopened (i.e., moved from the Closed to the Open list), else continue with (e)

    iii. if $s'$ is in the Open list as $s_o$ and $g(s') < g(s_o)$ then $s_o$ is updated (i.e., removed from the Open list and re-added in the next step with updated $g(\cdot)$), else continue with (e)

    iv. add $s'$ into the Open list

5. Walk back to retrieve the solution path.

In the above algorithm, $g(s)$ denotes a function assigning an accumulated cost $w$ for moving from the initial state $(s_0)$ to a given state $s$. During its execution, it always expands nodes with the lowest $f(s) = \alpha g(s) + \beta h(s)$. Different settings of $\alpha$ and $\beta$ give rise to different algorithms: for A*, $\alpha = \beta = 1$; for GBFS $\alpha = 0$ and $\beta = 1$.

*Consistent heuristics*, which are of special interest, are called monotone because the estimated cost of a partial solution $f(s) = g(s) + h(s)$ is monotonically non-decreasing along the best path to the goal. More than this, $f$ is monotone on all edges $(s, s')$, if and only if $h$ is consistent as we have $f(s') = g(s') + h(s') \geq g(s) + w(s, s') + h(s) - w(s, s') = f(s)$. For the case of consistent heuristics, no reopening (moving back nodes from Closed to Open) in A* is needed, as we essentially traverse a state-space graph with edge weights $w(s, s') + h(s') - h(s) \geq 0$. For the trivial heuristic $h_0$, we have $h_0(s) = 0$ and for perfect heuristic $h^*$, we have $f(s) = f^* = g(s) + h^*(s)$ for all nodes $s$. Both heuristics $h_0$ and $h^*$ are consistent. For GBFS and other variants, reopening is usually neglected. Even if the heuristic is not consistent, best-first algorithms without the reopening, remain complete i.e., they find a plan if there is one. Plans might not be provably optimal but are often good for planning practice [14].

## 3 Conditions on strictly optimally efficient heuristic

Let $\mathcal{O}_i$ be an Open list in the $i^{\text{th}}$ iteration of the forward search expanding only states on the optimal path $\pi$. Below definition defines a *perfect ranking heuristic* as a heuristic function where state on some optimal path in the Open list, $\mathcal{S}^{\pi:i} \cap \mathcal{O}_i$, has *always* strictly lower merit value than other states in the Open list off the optimal path, $\mathcal{O}_i \setminus \mathcal{S}^{\pi:i}$. The intersection $\mathcal{S}^{\pi:i} \cap \mathcal{O}_i$ contains always exactly one state.

**Definition 1** (Perfect ranking heuristic). *A heuristic function $h(s)$ is a **perfect ranking** in forward search with a merit function $f(s) = \alpha g(s) + \beta h(s)$ for a problem instance $\Gamma = (\langle \mathcal{S}, \mathcal{E}, w \rangle, s_0, \mathcal{S}^*)$ if and only if there exists an optimal plan $\pi = ((s_0, s_1), (s_1, s_2), \ldots, (s_{l-1}, s_l))$ such that*

- *$g(s)$ is the cost from $s_0$ to $s$ in a search-tree created by expanding only states on the optimal path $\pi$;*

- *$\forall i \in \{1, \ldots, l\}$ and $\forall s_j \in \mathcal{O}_i \setminus \mathcal{S}^{\pi:i}$ we have $f(s_j) > f(s_i)$.*

As an example consider a search-tree in Figure 1a with an optimal path $((s_0, s_1), (s_1, s_2), (s_2, s_3))$ and states $\{s_4, s_5, s_6, s_7\}$ off the optimal path. Then the Open list after expanding $s_0$ is $\mathcal{O}_1 = \{s_1, s_4, s_5\}$ and that after expanding $s_1$ is $\mathcal{O}_2 = \{s_2, s_4, s_5, s_6, s_7\}$. The set of states on the optimal sub-path are $\mathcal{S}^{\pi:1} = \{s_0, s_1\}$, $\mathcal{S}^{\pi:2} = \{s_0, s_1, s_2\}$. States on the optimal path in the Open lists are $\mathcal{S}^{\pi:1} \cap \mathcal{O}_1 = \{s_1\}$ and $\mathcal{S}^{\pi:2} \cap \mathcal{O}_2 = \{s_2\}$. States off the optimal path in the Open lists are $\mathcal{O}_1 \setminus \mathcal{S}^{\pi:1} = \{s_4, s_5\}$ and $\mathcal{O}_2 \setminus \mathcal{S}^{\pi:2} = \{s_4, s_5, s_6, s_7\}$. Function $f$ is perfectly ranking problem instance in Figure 1a iff following inequalities holds:

$$f(s_1) < f(s_4), \qquad f(s_2) < f(s_4), \qquad f(s_2) < f(s_6),$$
$$f(s_1) < f(s_5), \qquad f(s_2) < f(s_5), \qquad f(s_2) < f(s_7).$$

Notice that the heuristic values of states on the optimal path are never compared. This is because if the forward search expands only states on the optimal path, then its Open list always contains only

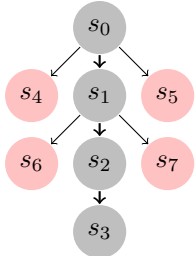
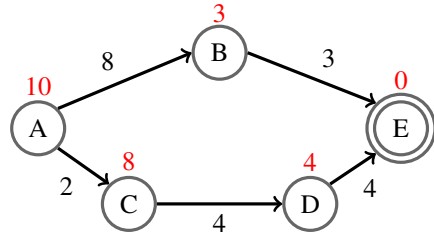

(a) An example of a search tree created by expanding only states on the optimal path $(s_0, s_1, s_2, s_3)$. Grey nodes denote states on the optimal path and pink nodes denote states off the optimal path.

(b) Problem instance where perfect heuristic is not strictly optimally efficient with GBFS. Numbers on the edges denote the cost of action and red numbers next to nodes denote the minimal cost-to-goal.

one state from the optimal path. The definition anticipates the presence of multiple optimal solutions and even multiple goals, which requires the heuristic to break ties and have a perfect rank with respect to one of them.

**Theorem 1.** *The forward search with a merit function $f(s) = \alpha g(s) + \beta h(s)$ and a heuristic $h$ is strictly optimally efficient on a problem instance $(\langle \mathcal{S}, \mathcal{E}, w \rangle, s_0, \mathcal{S}^*)$ if and only if $h$ is a perfect ranking on it.*

*Proof. Sufficiency:* If the conditions of a perfect ranking heuristic hold, then there exists an optimal path such that a state on it that is in the Open list always has the strictly lowest heuristic value. Therefore, forward search will never move off the optimal path and is, therefore, strictly optimally efficient.

*Necessity:* If $h$ is a strictly optimally efficient heuristic, then the forward search always selects the state on the optimal path, which means that the state has the lowest heuristic value of all the states in the Open list, which is precisely the condition of a perfect ranking heuristic. $\square$

Theorem 1, despite being trivial, allows certifying that the forward search with a given heuristic in a given problem instance is strictly optimally efficient. This property has been discussed in [59] for Beam search, but the conditions stated there are different because Beam search prunes states in the Open list. Recall that the complexity class of computing a perfect ranking heuristic function is the same as solving the problem because one implies the other. We now remind the reader that the popular cost-to-goal does not always lead to a strictly optimally efficient forward search.

**Example 1.** *While cost-to-goal $h^*$ is the best possible heuristic for algorithms like A\* (up to tie-breaking) in terms of nodes being expanded, for GBFS, $h^*$ does not necessarily yield optimal solutions.*

See Figure 2 for a counterexample. The complete proof is in the Appendix. Appendix 8.4 also gives an example of a problem instance, where an optimally efficient heuristic for GBFS returning the optimal solution does not exist.

## 4 Loss functions

### 4.1 Ranking loss function

Let's now design a loss function minimizing the number of violated conditions in Definition 1 for a problem instance $(\Gamma, s_0, \mathcal{S}^*)$ and its optimal plan $\pi$. Assuming a heuristic function $h(s, \theta)$ with parameters $\theta \in \Theta$, the number of violated conditions can be counted as

$$L_{01}(h, \Gamma, \pi) = \sum_{s_i \in \mathcal{S}^\pi} \sum_{s_j \in \mathcal{O}_i \setminus \mathcal{S}^{\pi:i}} [\![ r(s_i, s_j, \theta) > 0 ]\!], \tag{1}$$

where

$$r(s_i, s_j, \theta) = \alpha(g(s_i) - g(s_j)) + \beta(h(s_i, \theta) - h(s_j, \theta)), \tag{2}$$

$\mathcal{O}_i$, $\mathcal{S}^{\pi:i}$ as used in Definition 1, $[\![\cdot]\!]$ denotes Iverson bracket being one if the argument is true and zero otherwise, and $\alpha$ and $\beta$ are parameters controlling the type of search, as introduced above.

In imitation learning, we assume that we have a training set $\mathcal{T}^{\mathrm{trn}}$ consisting of tuples of problem instances and optimal plans $\mathcal{T}^{\mathrm{trn}} = \{(\langle \mathcal{S}_i, \mathcal{E}_i, w_i \rangle, s_{0,i}, \mathcal{S}_i^*, \pi_i)\}_{i=1}^n$. To optimize parameters $\theta$ of a heuristic function $h(s, \theta)$, we propose to minimize the number of wrongly ranked states over all the problem instances in the training set,

$$\arg\min_{\theta \in \Theta} \sum_{(\Gamma, \pi) \in \mathcal{T}^{\mathrm{trn}}} \mathrm{L}_{01}(h, \Gamma, \pi). \tag{3}$$

In practice, one often wants to solve problem (3) by efficient gradient optimization methods. To do so, the Iverson bracket $[\![\cdot]\!]$ (also called 0-1 loss) is usually replaced by a convex surrogate such as the hinge-loss or the logistic loss used in the Experimental section below, in which case Equation (3) becomes

$$\arg\min_{\theta \in \Theta} \sum_{(\Gamma, \pi) \in \mathcal{T}^{\mathrm{trn}}} \sum_{s_i \in \mathcal{S}^\pi} \sum_{s_j \in \bar{\mathcal{S}}^i} \log(1 + \exp(r(s_i, s_j, \theta))). \tag{4}$$

While optimization of the surrogate loss might look as if one optimizes a different problem, it has been shown that for certain classes of functions, optimization of surrogate losses solves the original problem (3) as the number of training samples approach infinity [50, 35] .

## 4.2 Regression loss function

For the sake of comparison, we review a loss function used in the majority of works optimizing the cost-to-goal in imitation learning for planning (some notable exceptions are in the related work section). The optimization problem with $\mathrm{L}_2$ loss function is defined as

$$\arg\min_{\theta \in \Theta} \sum_{(\Gamma, \pi) \in \mathcal{T}^{\mathrm{trn}}} \sum_{s_i \in \mathcal{S}^\pi} (h(s_i, \theta) - h^*(s_i))^2, \tag{5}$$

where $h^*(s_i)$ is the true cost-to-goal. The optimal solution of the optimization is the heuristic function $h^*$. What is important for the discussion below is that the $\mathrm{L}_2$ loss function uses *only states on the optimal path* and the heuristic function is optimized to solve the *regression* problem. Other variants of regression loss, such as asymmetric $\mathrm{L}_1$ in [51], do not solve issues of regression losses discussed in this paper.

## 4.3 Advantages and disadvantages of loss functions

**Cost-to-goal provides a false sense of optimality**  It has been already mentioned above that A* with $h^*$ is strictly optimally efficient up to resolving states with the same value of $g(s) + h^*(s)$. This means that for problems with a large (even exponential) number of optimal solutions of equal cost (see [23] for examples), A* will be very inefficient unless some mechanism for breaking ties is used. Moreover, since learning is inherently imprecise, a heuristic close but not equal to $h^*$ will likely be less efficient than the one close to a perfect-ranking heuristic.

**Size of the solution set**  The set of all functions satisfying Equation (5) is likely smaller than that satisfying Equation (3), because the solution set of (3) is invariant to transformations by *strictly monotone* functions, unlike the solution set of (5). From that, we conjecture that finding the solution of (5) is more difficult. The precise theorem is difficult to prove since the solution of (5) is not a solution of (3) due to tie resolution in Definition 1.

**Optimizing cost-to-goal does not utilize states off the solution path**  Assume a problem instance $(\langle \mathcal{S}, \mathcal{E}, w \rangle, s_0, \mathcal{S}^*)$ and an optimal solution path $\pi = ((s_0, s_1), (s_1, s_2), \ldots, (s_{l-1}, s_l))$. Loss function optimized in (5) uses only states $\{s_i\}_{i=0}^l$ on the optimal path. This means that even when the loss is zero, the search can be arbitrarily inefficient if states $\mathcal{S} \backslash \mathcal{S}^\pi$ off the optimal path will have heuristic values smaller than $\min_{s \in \mathcal{S}^\pi} h(s)$. The proposed ranking loss (3) uses states on and off the optimal path and therefore, does not suffer from this problem. This issue can be fixed if the training set is extended to contain heuristic values for all states off the optimal path (states $\{s_4, s_5, s_6, s_7\}$ in Figure 1a), as has been suggested in [5, 49]. With respect to the above definitions, this is equal to adding more problem instances to the training set sharing the same graph $\Gamma$ and set of goal states $\mathcal{S}^*$, but differing in initial states. Solving all of them to obtain true cost-to-goal *greatly increases the cost of creating the training set*.

**Heuristic value for dead-end states**    should be sufficiently high to ensure they are never selected. An infinity in the $L_2$ loss would always lead to an infinite gradient in optimization. In practice, the $\infty$ is replaced by a sufficiently large value, but too large values can cause large gradients of $L_2$, which might negatively impact the stability of convergence of gradient-based optimization methods. Proposed ranking losses do not suffer this problem.

**Goal-awareness**    The drawback of the heuristic function $h(s, \theta)$ obtained by optimizing ranking losses is that they are not goal-aware unlike the heuristic optimizing the estimate of cost-to-goal.

**Speed of convergence against the size of training set**    From classical bounds on true error [54] and [7], we can derive (see Appendix for details) that the excess error of ranking loss converges to zero at a rate $\frac{\sqrt{\ln n_p}}{\sqrt{n_p}}$, which is slightly faster than that of regression $\frac{\sqrt{\ln n_s}}{\sqrt{n_s} - \sqrt{\ln n_s}}$, where $n_s$ and $n_p$ denotes the number of states and state pairs respectively in the training set. But, the number of state pairs in the training set grows quadratically with the number of states; therefore, the excess error of ranking loss for the number of states $n_s$ can be expressed as $\frac{\sqrt{2 \ln n_s}}{n_s}$, which would be by at least $\frac{1}{\sqrt{n_s}}$ factor faster than that of regression. Note though that these rates are illustrative since the independency of samples (and sample pairs) is in practice, violated since even samples from the same problem instance are not independent.

**Conflicts in training set**    Unlike regression loss, the proposed family of ranking losses is potentially sensitive to conflicts in the training set, which occur when the training set contains examples of two (or more) different solution paths of *the same* problem instance. This might prevent achieving a zero loss. The effect on the heuristic depends on the composition of the training set. Either solution paths more frequent in the training set will be preferred, or there will be ties, but their resolution do not affect optimally efficiency. Appendix 8.5 provides an illustrative example.

## 5    Related Work

The closest work to ours is [36, 38], which adjusts heuristic value by a policy estimated from the neural network. The theory therein has the same motivation, to minimize the number of expanded states, though it is more complicated than the theory presented here. For efficiency, [38] needs to modify the GBFS such that search nodes store policy and probability of reaching the node from the start. Moreover, in some variants (used in the experimental comparison below) the neural network has to estimate both heuristic value and policy.

Inspired by statistical surveys of heuristic functions in [57, 58], [18] proposes to optimize the rank of states preserving order defined by the true cost-to-goal. The formulation is therefore valid only for GBFS and not for A* search. Ties are not resolved, which means the strict optimal efficiency cannot be guaranteed. Similarly, heuristic values for domains where all actions have zero cost cannot be optimized. Lastly and importantly, the need to know the true cost-to-goal for all states in the training set greatly increases the cost of its construction as discussed above. From Bellman's equation [49] arrives at the requirement that the smallest heuristic value of child nodes has to be smaller than that of the parent node. The resulting loss is regularized with $L_1$ forcing the value of the heuristic to be close to cost-to-goal.

In [5], A* is viewed as a Markov Decision Process with the Q-function equal to the number of steps of A* reaching the solution. This detaches the heuristic values from the cost-to-goal cost, but it does not solve the problem with dead-ends and ties, since the optimization is done by minimizing $L_2$ loss.

Symmetry of $L_2$ (and of $L_1$) loss are discussed in [51], as it does not promote the admissibility of the heuristic. It suggests asymmetric $L_1$ loss with different weights on the left and right parts, but this does not completely solve the problem of estimating cost-to-goal (for example with ties, dead-ends).

Parameters of potential heuristics [41] are optimized by linear programming for each problem instance to meet admissibility constraints while helping the search. On contrary, this work focuses on the efficiency of the search for many problem instances but cannot guarantee admissibility in general.

Combining neural networks with discrete search algorithms, such that they become an inseparable part of the architecture is proposed in [55] and [60]. The gradient with respect to parameters has to be

propagated through the search algorithm which usually requires both approximation and the search algorithm to be executed during every inference.

A large body of literature [46, 20, 15, 2] improves the Monte Carlo Tree Search (MCTS), which has the advantage to suppress the noise in estimates of the solution costs. These methods proposed there are exclusive to MCTS and do not apply to variants of the forward search algorithm. Similarly, a lot of works [45, 53, 61, 19, 8, 16, 5] investigate architectures of neural networks implementing the heuristic function $h(s, \theta)$, ideally for arbitrary planning problems. We use [45, 8] to implement $h(s, \theta)$, but remind our readers that our interest is the loss function minimized when optimizing $\theta$.

Finally, [1] proposes to construct a training set similarly to pattern databases [12] by random backward movements from the goal state and to adapt A* for multi-core processors by expanding $k > 1$ states in each iteration. These improvements are independent of those proposed here and can be combined.

## 6 Experiments

The experimental protocol was designed to compare the loss functions while removing all sources of variability. It therefore uses imitation learning in planning [18, 19] and leaves more interesting bootstrap protocol for future work [3]. The goal was not to deliver state-of-the-art results, therefore the classical solvers were omitted, but their results are in the Appendix for completeness.

The proposed ranking losses are instantiated with the logistic loss function as in Equation (4) for A* search by setting $\alpha = \beta = 1$ in (2), called L*, and for GBFS, by setting $\alpha = 0$, $\beta = 1$ in (2), called $L_{\mathrm{gbfs}}$. $L_2$ is used as defined in Equation (5). Since the heuristics used in this paper are not admissible, A* search will not provide optimal solutions. It was included in the comparison to demonstrate that heuristics optimized with respect to $L_{\mathrm{gbfs}}$ might be inferior in A* search. To observe the advantage of ranking over regression, we defined a $L_{\mathrm{rt}}$ loss function

$$L_{\mathrm{rt}}(h, \Gamma, \pi) = \sum_{(s_{i-1}, s_i) \in \pi} \log(1 + \exp(h(s_i, \theta) - h(s_{i-1}, \theta))) \tag{6}$$

comparing only states on the optimal trajectory. We also compare to the loss function proposed in [49] defined as $L_{\mathrm{be}}(h, \Gamma, \pi) = \sum_{s \in \mathcal{S}^\pi} \max\{1 + \min_{s' \in \mathcal{N}(s)} h(s', \theta) - h(s, \theta), 0\} + \max\{0, h^*(s) - h(s, \theta)\} + \max .\{0, h(s, \theta) - 2h^*(s)\}$, where $\mathcal{N}(s)$ denotes child-nodes of state $s$. Furthermore, we compare to the policy-guided heuristic search [38] (denoted as $L_{\mathrm{le}}$) with GBFS modified for efficiency as described in the paper while simultaneously adapting the neural network to provide the policy and heuristic value.

The heuristic function $h(s, \theta)$ for a given problem domain was optimized on the training set containing problem instances and their solutions, obtained by SymBA* solver [52] for Sokoban and Maze with teleports and [42] [1] for Sliding Tile. Each minibatch in SGD contained states from exactly a single problem instance needed to calculate the loss. This means that for $L_2$, $L_{\mathrm{rt}}$, $L_{\mathrm{le}}$, only states on the solution trajectory were included; for L*, $L_{\mathrm{gbfs}}$, and $L_{\mathrm{be}}$ there were additional states of distance one (measured by the number of actions) from states on the solution trajectory. Optimized heuristic functions were always evaluated within A* and GBFS forward searches, such that we can analyze how the heuristic functions specialize for a given search. To demonstrate the generality, experiments contained eight problem domains of two types (grid and general PDDL) using two different architectures of neural networks described below.

**Neural network for grid domains**    States in *Sokoban*, *Maze* with teleports, and *Sliding-puzzle* can be easily represented in a grid (tensor). The neural network implementing the heuristic function $h(s, \theta)$ was copied from [8]. Seven convolution layers, $P_1 \ldots P_7$ of the same shape $3 \times 3$ with 64 filters are followed by four CoAT blocks (CoAT block contains a convolution layer followed by a multi-head attention operation with 2 heads and a positional encoding layer). Convolution layers in CoAT blocks have size $3 \times 3$ and have 180 filters. The network to this stage preserves the dimension of the input. The output from the last block is flattened by mean pooling along $x$ and $y$ coordinates before being passed onto a fully connected layer (FC) predicting the heuristic. All blocks have skip connections.

---

[1]runs the "LAMA 2011 configuration" of the planner

**Neural network for PDDL domains**  For *Blocksworld*, *Ferry*, *Spanner*, *N-Puzzle*, and *Elevators*, where the state cannot be easily represented as a tensor, we have used Strips-HGN [44] and we refer the reader therein for details. Strips-HGN was implemented in Julia [4] using `PDDL.jl` [62] extending [30]. Importantly, we have used the hyper-graph reduplication proposed in [48] to improve computational efficiency. Since we did not know the precise architecture (unlike in the grid domain), we performed a grid-search over the number of hyper-graph convolutions in $\{1, 2, 3\}$, dimensions of filters in $\{4, 8, 16\}$, and the use of residual connections between features of vertices. The representation of vertices was reduced by mean and maximum pooling and passed to a two-layer feed-forward layer with hidden dimension equal to the number of filters in hyper-graph convolutions. All non-linearities were of relu type. The best configuration was selected according to the number of solved problems in the validation set. Forward search algorithms were given 5s to solve each problem instance chosen in a manner to emphasize differences between loss functions. All experiments were repeated 3 times. All experiments on the PDDL domains including development took 17724 CPU hours. Optimizing heuristic for a single problem instance took approximately 0.5 CPU/h except Gripper domain, where the training took about 8h due to large branch factors. The rest of this subsection gives further details about domains.

**Sokoban**  The training set for the Sokoban domain contained 20000 mazes of grid size $10 \times 10$ with a single agent and 3 boxes. The testing set contained 200 mazes of the same size $10 \times 10$, but with 3, 4, 5, 6, 7 boxes. Assuming the complexity of to be correlated with the number of boxes, we can study generalization to more difficult problems. The problem instances were generated by [40].

**Maze with teleports**  The training set contained 20000 mazes of size $15 \times 15$ with the agent in the upper-left corner and the goal in the lower-right corner. The testing set contained 200 mazes of size $50 \times 50$, $55 \times 55$, and $60 \times 60$. Therefore, as in the case of Sokoban, the testing set was constructed such that it evaluates the generalization of the heuristic to bigger mazes. The mazes were generated using an open-source maze generator [47], where walls were randomly broken and 4 pairs of teleports were randomly added to the maze structure.

**Sliding puzzle**  The training set contained 20000 puzzles of size $5 \times 5$. The testing set contained 200 mazes of size $5 \times 5$. These puzzles were taken from [37]. Furthermore, we tested our approach on puzzles of higher dimensions such $6 \times 6$ and $7 \times 7$, all of which were generated with [37]. Thus, in total, the testing set contained 200 sliding tile puzzles for each of the three dimensions.

**Blocksworld, N-Puzzle, Ferry, Elevators, and Spanner**  The problem instances were copied from [44] (Blocksworld, N-Puzzle, and Ferry) and from [34] (`elevators-00-strips`). Problem instances of Spanner were generated by [43]. Blocksworld contained 732 problem instances of size 3–15, N-Puzzle contained 80, Ferry contained 48, Elevators contained 80, and Spanner contained 440. Table 5 in Appendix shows the fraction of -olved mazes by breadth-first search for calibration of the difficulty. The problem instances were randomly divided into training, validation, and testing sets, each containing 50% / 25% / 25% of the whole set of problems. The source code of all experiments is available at https://github.com/aicenter/Optimize-Planning-Heuristics-to-Rank with MIT license.

## 6.1   Experimental results

Table 1 shows the fraction of solved mazes in percent for all combinations of search algorithms (A* and GBFS) and heuristic functions optimized against compared loss functions. The results match the above theory, as the proposed ranking losses are *always better* than the $L_2$ regression loss.

For A* search, the heuristic function optimized against the $L^*$ is clearly the best, as it is only once inferior to $L_{rt}$ and $L_{be}$ on the Ferry domain. As expected, heuristic functions optimized with respect to $L_{gbfs}$ behave erratically in A* search, because they were optimized for a very different search.

For GBFS, heuristic functions optimized against the $L^*$ or against $L_{gbfs}$ behave the best. It can be shown (the proof is provided in Apendix) that since each action has a constant and positive cost, the heuristic function optimizing $L^*$ optimizes $L_{gbfs}$ as well, but the opposite is not true.

Heuristic functions optimized against $L_2$ estimating cost-to-goal deliver most of the time the worst results. Heuristic functions optimized against other ranking losses $L_{rt}$ and $L_{be}$ frequently perform well, most of the time better than $L_2$, but sometimes (Elevators, Sokoban, Sliding puzzle) are much

| problem | complx. | A* | | | | | GBFS | | | | | |
|---|---|---|---|---|---|---|---|---|---|---|---|---|
| | | $L^*$ | $L_{gbfs}$ | $L_{rt}$ | $L_2$ | $L_{be}$ | $L^*$ | $L_{gbfs}$ | $L_{rt}$ | $L_2$ | $L_{be}$ | $L_{le}$ |
| Blocks | | **100** | **100** | **100** | 99 | **100** | **100** | **100** | **100** | **100** | **100** | 99 |
| Ferry | | 98 | 98 | **100** | 92 | **100** | 98 | **100** | **100** | **100** | 98 | 98 |
| N-Puzzle | | **89** | 87 | 88 | 83 | **89** | **92** | 89 | 89 | 89 | **92** | 88 |
| Spanner | | **100** | 89 | **100** | 84 | 92 | **100** | **100** | **100** | **100** | **100** | **100** |
| Elevators | | **91** | 85 | 75 | 36 | 66 | **92** | 85 | 79 | 76 | 67 | 58 |
| Sokoban | 3 boxes | **99** | 98 | 96 | 97 | 92 | 98 | **100** | 94 | 95 | 92 | 98 |
| | 4 boxes | **89** | 89 | 85 | 81 | 82 | 87 | **91** | 84 | 83 | 84 | 84 |
| | 5 boxes | **80** | 75 | 72 | 72 | 73 | **78** | 77 | 74 | 72 | 72 | 73 |
| | 6 boxes | **76** | 69 | 59 | 51 | 53 | **73** | 71 | 56 | 51 | 54 | 64 |
| | 7 boxes | **55** | 49 | 47 | 42 | 45 | **51** | 49 | 48 | 43 | 45 | 49 |
| Maze w. t. | $50 \times 50$ | **92** | 91 | 88 | 87 | 87 | 89 | **90** | 89 | 84 | 85 | 89 |
| | $55 \times 55$ | **78** | 75 | 73 | 72 | 74 | 74 | **75** | 74 | 72 | **75** | 74 |
| | $60 \times 60$ | **49** | 37 | 35 | 32 | 31 | 42 | **48** | 36 | 34 | 32 | 42 |
| Sliding puzzle | $5 \times 5$ | **88** | 83 | 84 | 80 | 82 | 86 | **87** | 84 | 84 | 84 | 85 |
| | $6 \times 6$ | **51** | 48 | 49 | 45 | 46 | 47 | **49** | 45 | 43 | 46 | 48 |
| | $7 \times 7$ | **39** | 35 | 36 | 32 | 34 | 35 | **36** | 35 | 32 | 34 | 35 |

Table 1: Fraction of solved problem instances in percent from the testing set. The top row denotes the type of the search and the second top row denotes the loss function against which the heuristic function was optimized. The complexity is explicitly shown for grid domains. Standard deviations are most of the times smaller than one percent and are provided in Table in the Appendix.

worse than proposed losses. These experiments further confirm the [57, 18, 59] stating that optimizing the exact estimate of cost-to-goal is unnecessary, but they also show the advantage of including states off the optimal path in the loss function. Finally, the method combining policy and heuristic in GBFS, $L_{le}$, [38] performs better that $L_2$, but worse than the proposed ranking losses. Since both approaches are derived from the same motivation, we attribute this to difficulties in training neural networks with two heads. Ranking losses optimize just the heuristic which is directly used in the search, which seems to us to be simpler. Additional statistics, namely the average number of expanded states (Table 6) and the average length of plans (Table 8) are provided in the Appendix.

## 7    Conclusion

The motivation of this paper stemmed from the observation that even the cost-to-goal, considered to be an optimal heuristic, fails to guarantee a strictly optimally efficient search. Since a large body of existing research optimizes this quantity, we are effectively lost with respect to what should be optimized. To fill this gap, we have stated the necessary and sufficient conditions guaranteeing the forward search to be strictly optimally efficient. These conditions show that the absolute value of the heuristic is not important, but that the ranking of states in the Open list is what controls the efficiency. Ranking can be optimized by minimizing the ranking loss functions, but its concrete implementation needs to correspond to a variant of the forward search. In case of mismatch, the resulting heuristic can perform poorly, which has been demonstrated when the heuristic optimized for BGFS search was used with A* search. The other benefit of ranking losses is that from the point of view of statistical learning theory, they solve a simpler problem than ubiquitous regression in estimating the cost-to-goal.

The experimental comparisons on eight problem domains convincingly support the derived theory. Heuristic functions optimized with respect to ranking loss $L^*$ instantiated for A* search perform almost always the best with A* search (except for one case where it was the second best). In the case of GBFS, heuristic functions optimizing ranking losses $L_{gbfs}$ and $L^*$ instantiated for GBFS and A* worked the best. This is caused by the fact that the heuristic optimizing $L^*$ for A* optimizes $L_{gbfs}$ for GBFS as well.

We do not question the existence of a strictly optimally efficient heuristic. Given our experiments, we believe that if the space of heuristic functions over which the loss is optimized is sufficiently rich, the result will be sufficiently close to the optimal for the needs of the search.

### 7.1 Acknowledgements

This work has been supported by project numbers 22-32620S and 22-30043S from Czech Science Foundation and OP VVV project CZ.02.1.01/0.0/0.0/16_019/0000765 "Research Center for Informatics". This work has also received funding from the European Union's Horizon Europe Research and Innovation program under the grant agreement TUPLES No 101070149.

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

# 8 Appendix

## 8.1 Speed of convergence against the size of training set:

We use the classical results of statistical learning theory to show that estimating the cost-to-go converges more slowly with respect to the size of the training set than estimating the rank.

From Equation (1) in the main text it should be obvious that ranking is in its essence a classification problem if one considers a pair of states $(s, s')$ as a single sample. For classification/ranking problems, a following bound on true error rate [54] holds with probability $1 - \eta$

$$R_c \leq \hat{R}_c + \sqrt{\frac{1}{n_p} \left[ \left( 1 + \kappa \ln \frac{2n_p}{\kappa} \right) - \ln \eta \right]},$$

where $R_c$ denotes the true loss (Equation (1) in the main text) and $\hat{R}_c$, its estimate from $n_p$ state pairs $(s_i, s_j)$, and $\kappa$, the Vapnik-Chervonenkis dimension [54] of the hypothesis space.

Optimizing $h(s, \theta)$ with respect to cost-to-goal (Equation (5) in the main text) is a regression problem for which a different generalization bound on prediction error [7] holds with probability $1 - \eta$

$$R_r \leq \hat{R}_r \left[ 1 - \sqrt{\frac{1}{n_s} \left[ \kappa \left( 1 + \ln \frac{n_s}{\kappa} \right) - \ln \eta \right]} \right]_+^{-1}$$

where again $R_r$ is the error of the estimator of cost-to-go and $\hat{R}_r$ is its estimate from $n_s$ states (Equation (5) in the main text),[2] and $[x]_+$ is a shorthand for $\max\{0, x\}$.

From the above, we can see that excess error in the ranking case converges to zero at a rate $\frac{\sqrt{\ln n_p}}{\sqrt{n_p}}$, which is slightly faster than that of regression $\frac{\sqrt{\ln n_s}}{\sqrt{n_s} - \sqrt{\ln n_s}}$. But, the number of state pairs in the training set grows quadratically with the number of states; therefore, the convergence rate of the ranking problem for the number of states $n_s$ can be expressed as $\frac{\sqrt{2 \ln n_s}}{n_s}$, which would be by at least $\frac{1}{\sqrt{n_s}}$ factor faster than that of regression. We note that bounds are illustrative, since the independency of samples is in practice violated, since samples from the same problem-instance are not independent.

## 8.2 Suboptimality of perfect heuristic in GBFS

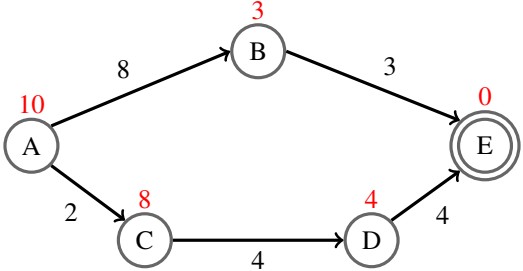

Figure 2: Problem instance where perfect heuristic is not strictly optimally efficient with GBFS. Numbers on the edges denote the cost of action and red numbers next to nodes denote the minimal cost-to-go.

**Example 2.** *While cost-to-goal $h^*$ is the best possible heuristic for algorithms like A* (up to tie-breaking) in terms of nodes being expanded, for GBFS, $h^*$ does not necessarily yield optimal solutions.*

*Proof.* A* explores all the nodes with $f(s) < f^*(s) = g(s) + h^*(s)$ and some with $f(s) = f^*(s) = g(s) + h^*(s)$, so nodes can only be saved with optimal $f^*(s) = g(s) + h^*(s)$. In fact, when given

---

[2]The formulation in [7] contains constants $c$ and $a$, but authors argue they can be set to $a = c = 1$ and hasubve been therefore omitted here.

$h^*$ as the heuristic, *only* nodes $s$ with $f(s) = f^*(s)$ are expanded. Depending on the strategy of tie-breaking, the solution path can be found in the minimal number of node expansions or take significantly longer (e.g., in lower $g$-value first exploration of the search frontier). Any heuristic other than $h^*$ is either overestimating, and, therefore, may lead to either non-optimal solutions in A\*, or weaker than $h^*$, leading to more nodes being expanded.

Even if $h^*$ is given, GBFS is not guaranteed to be optimal. Consider the following graph with five nodes $A, B, C, D, E$, and $w(A, B) = 8, w(B, E) = 3, w(A, C) = 2, w(C, D) = 4, w(D, E) = 4$, and $h^*(A) = 10, h^*(B) = 3, h^*(C) = 8, h^*(D) = 4, h^*(E) = 0$ (see Figure 2), initial node $s_0 = A$, goal node $E \in \mathcal{S}^*$. The numbers are the actual costs and the red numbers are the exact heuristic function. For finding a path from node $A$ to node $E$, GBFS would return $(A, B, E)$ following the heuristic function. However, the path $(A, C, D, E)$ has cost 10 instead of 11. $\square$

## 8.3 Theory

Below, we prove the claim in made in Experimental section stating that if a heuristic $h$ is strictly optimally efficient for A\* search, then it is also strictly optimally efficient for GBFS.

**Theorem 2.** *Let a heuristic $h$ is a perfect ranking for A\* search on a problem instance $\Gamma = (\langle \mathcal{S}, \mathcal{E}, w \rangle, s_0, \mathcal{S}^*)$ with a constant non-negative cost of actions $((\exists c \geq 0) (\forall e \in \mathcal{E}) (w(e) = c))$. Then $h$ is a perfect ranking for GBFS on $\Gamma$.*

*Proof.* Let $\pi = ((s_0, s_1), (s_1, s_2), \ldots, (s_{l-1}, s_l))$ be an optimal plan such that $\forall i \in \{1, \ldots, l\}$ and $\forall s_j \in \{s_j \mid \exists (s_k, s_j) \in \mathcal{E} \wedge s_k \in \mathcal{S}^{\pi:i-1} \wedge s_j \notin \mathcal{S}^{\pi:i}\}$ we have $g(s_j) + h(s_j) > g(s_i) + h(s_i)$, where $g(s)$ is the distance from $s_0$ to $s$ in a search-tree created by expanding only states on the optimal path $\pi$. We want to proof that if all actions have the same positive costs, then $h(s_j) > h(s_i)$ as well.

We carry the proof by induction with respect to the number of expanded states.

At the initialization (step 0) the claim trivially holds as the Open list contains just a root node and the set of inequalities is empty.

Let's now make the induction step and assume the theorem holds for the first $i - 1$ step. We divide the proof to two parts. At first, we prove the claim for $(O)_i \setminus \mathcal{N}(s_{i-1})$ and then we proof the claim for $\mathcal{N}(s_{i-1})$, where $\mathcal{N}(s)$ denotes child states of the state $s$.

1) Assume $s_j \in (O)_i \setminus \mathcal{N}(s_{i-1})$. Since $h$ is strictly optimally efficient for A\*, it holds that

$$\begin{aligned} g(s_j) + h(s_j) &> g(s_i) + h(s_i) \\ h(s_j) &> (g(s_i) - g(s_j)) + h(s_i) \\ h(s_j) &> h(s_i), \end{aligned}$$

where the last inequality is true because $g(s_i) - g(s_j) \geq 0$.

Assume $(s_j \in \mathcal{N}(s_{i-1}))(s_j \neq s_i)$. Since $h$ is strictly optimall efficient for A\*, it holds that

$$g(s_j) + h(s_j) > g(s_i) + h(s_i). \tag{7}$$

Since $g(s_j) = w((s_{i-1}, s_j)) = w((s_{i-1}, s_i)) = g(s_i)$, it holds

$$h(s_j) > h(s_i), \tag{8}$$

which finishes the proof of the theorem. $\square$

## 8.4 Optimally efficient heuristic might not exists for GBFS

Consider the following graph in Figure 3 with four nodes $A, B, C, D,$, and $w(A, B) = 1, w(B, D) = 1, w(A, C) = 1, w(A, D) = 9, w(B, C) = 9$, and $h^*(A) = 0, h^*(B) = 1, h^*(C) = 1, h^*(D) = 2$ where A is the goal state and D is the initial state.

We can see that for GBFS, the perfect heuristic does not exist for D. On expansion, the GBFS algorithm will put A and B to the open list with heuristic values h(A) = 0 and h(B) = 1. GBFS takes A from the open list and checks if it is a goal state. Since A is goal state, GBFS terminates returning

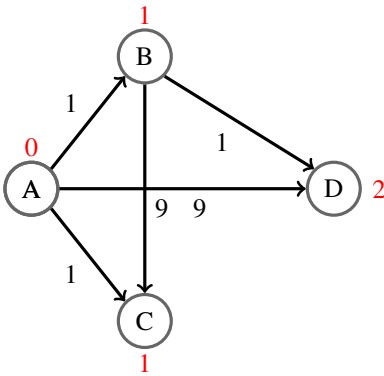

Figure 3: Problem instance where optimally efficient heuristic does not exists for GBFS.

path (D,A) as a solution. However, this is not the optimal path as a better (optimal) solution exists (D, B, A). Since the definition of optimal ranking requires the inequalities to hold for this optimal path, in GBFS, the perfect heuristic does not exist for all initial states.

The problem can be fixed if the definition of optimal ranking is changed to consider two cases in the merit function $f(s) = \alpha g(s) + \beta h(s)$: $\alpha > 0$ and $\beta > 0$ and $\alpha = 0$ and $\beta > 0$. In the first case, the optimal ranking should be defined with respect to the "optimal path" (this is the A*); in the latter case, it should be the path with minimal number of expansions. With GBFS, the user simply wants to find a solution but not care about its optimality. With this change, the heuristic function will exist for Figure 3.

## 8.5 Training set with multiple solution paths with the same length of the plan

The behavior of the learned heuristic function depends on the composition of the training set, which is illustrated below on a simple grid problem. The agent starts at position (4,4) and has to move to the goal position (0,0). There are no obstacles and the agent can only move one tile down or one tile left (the effects on his positions are either (-1,0) or (0,-1)), the cost of the action is one. The number of different solutions path grows exponentially with the size of the grid and all solution path has the same cost (for problem of size 5x5, there are 70 solutions). This problem is interesting, since merit function in A* with optimal heuristic function (cost to goal) is constant, equal to 8, for all states.

For the size of the grid (4,4), the heuristic is determined by a table with 25 values. Below, these values are determined by minimizing the proposed loss for A* algorithm with logistic loss surrogate by BFGS (with all zeros as initial solution). Since the loss function is convex, BFGS finds the solution quickly.

In the first case, the training set contains one solution path ([4, 4], [3, 4], [2, 4], [1, 4], [0, 4], [0, 3], [0, 2], [0, 1], [0, 0]), where the agent first goes left and then down. The learnt heuristic values h is shown in Table 2.

| y/x | 0 | 1 | 2 | 3 | 4 |
|-----|------|------|------|------|------|
| 4 | -191.53 | -131.99 | -76.94 | -31.25 | 0.0 |
| 3 | 0.0 | 31.25 | 76.94 | 131.99 | 191.53 |
| 2 | 0.0 | 0.0 | 0.0 | 0.0 | 0.0 |
| 1 | 0.0 | 0.0 | 0.0 | 0.0 | 0.0 |
| 0 | 0.0 | 0.0 | 0.0 | 0.0 | 0.0 |

Table 2: Table showing one solution path.

An A* search using these heuristic values will first always take action moving the agent to left and then down. When moving down, the agent does not have another choice. Notice that the heuristic values of many states are not affected by the optimization, because they are not needed to effectively solve the problem.

In the second case, the training set contains two solution paths: the first is in Table 2 and in the second table 3, the agent goes first down and then left. The learnt heuristic values are shown in 3.

| y/x | 0 | 1 | 2 | 3 | 4 |
|---|---|---|---|---|---|
| 4 | -95.76 | -66.0 | -38.47 | 80.14 | 0.0 |
| 3 | 0.0 | 15.62 | 38.47 | 131.99 | 80.14 |
| 2 | 0.0 | 0.0 | 0.0 | 38.47 | -38.47 |
| 1 | 0.0 | 0.0 | 0.0 | 15.62 | -66.0 |
| 0 | 0.0 | 0.0 | 0.0 | 0.0 | -95.76 |

Table 3: Table showing two solution paths.

An A* search will now face tie at state (4,4), since states (3,4) and (4,3) have the same heuristic value. But the presence of the tie does not affect the optimal efficiency of the search, as A* will always expand states on one of the optimal path from the training set.

Finally let's consider a case, where all 70 possible solutions are in the training set. The learned heuristic values are shown in Table 4.

| y/x | 0 | 1 | 2 | 3 | 4 |
|---|---|---|---|---|---|
| 4 | 3.74 | 92.62 | 134.66 | 163.17 | 0.0 |
| 3 | -46.61 | 2.35 | 91.93 | 134.38 | 163.17 |
| 2 | -167.96 | -47.7 | 1.94 | 91.93 | 134.66 |
| 1 | -210.03 | -168.66 | -47.7 | 2.35 | 92.62 |
| 0 | 0.0 | -210.03 | -167.96 | -46.61 | 3.74 |

Table 4: Table showing multiple solution paths.

Rank of heuristic values "roughly" corresponds to cost to goal. The reason, why some states are preferred over the others despite their true cost-to-goal being the same is that they appear in more solution paths. As shown in Table 4, the A* search with learnt heuristic values is strictly optimally efficient.

## 8.6 Baseline Comparison to breadth-first search

The fraction of solved problems for breadth-first search (5s time limit as used by solvers in the paper) is shown in Table 5.

| domain | fraction |
|---|---|
| blocks | 0.35 |
| ferry | 0.31 |
| npuzzle | 0.14 |
| spanner | 0.63 |
| elevators | 0.32 |

Table 5: Fraction of solved mazes by breadth-first search.

## 8.7 Training Details

For the grid domains, ADAM [26] training algorithm was run for $100 \times 20000$ steps for the grid domains.The experiments were conducted in the Keras-2.4.3 framework with Tensorflow-2.3.1 as the backend. While all solvers were always executed on the CPU, the training used an NVIDIA Tesla GPU model V100-SXM2-32GB. Forward search algorithms were given 10 mins to solve each problem instance.

For the PDDL domains, the training consumed approximately 100 GPU hours and evaluation consumed 1000 CPU hours.All training and evaluation were done on single-core Intel Xeon Silver 4110 CPU 2.10GHz with a memory limit of 128GB. The training algorithm AdaBelief [63] was

allowed to do 10000 steps on the CPU. We emphasize though, that the training time does not include the cost of creating the training set.

| problem | complx. | A* | | | | | GBFS | | | | | |
|---|---|---|---|---|---|---|---|---|---|---|---|---|
| | | $L^*$ | $L_{gbfs}$ | $L_{rt}$ | $L_2$ | $L_{be}$ | $L^*$ | $L_{gbfs}$ | $L_{rt}$ | $L_2$ | $L_{be}$ | $L_{le}$ |
| blocks | | 37 | 54 | **27** | 137 | 54 | 33 | **28** | 29 | 32 | 32 | 127 |
| ferry | | 53 | 43 | 36 | 339 | **20** | 48 | 34 | 31 | 33 | **20** | 51 |
| npuzzle | | **294** | 660 | 843 | 1936 | 641 | 297 | 311 | 333 | 591 | **272** | 418 |
| spanner | | 55 | 546 | **53** | 807 | 416 | 61 | 56 | **53** | 117 | 65 | 148 |
| elevators | | **33** | 35 | 52 | 657 | 310 | **33** | **33** | 43 | 115 | 198 | 73 |
| Sokoban | 3 boxes | **14** | **14** | **14** | 17 | **14** | **14** | **14** | **14** | 17 | **14** | **14** |
| | 4 boxes | **32** | 35 | 37 | 44 | 35 | 34 | **32** | 36 | 43 | 35 | 33 |
| | 5 boxes | **61** | 67 | 68 | 72 | 63 | 65 | 62 | 66 | 77 | 64 | **61** |
| | 6 boxes | **171** | 179 | 180 | 210 | 177 | 175 | 179 | 181 | 214 | 180 | **174** |
| | 7 boxes | **643** | 651 | 653 | 755 | 654 | 645 | 641 | **640** | 754 | 655 | 643 |
| Maze w. t. | $50 \times 50$ | **34** | 37 | **34** | 41 | 40 | 34 | **33** | 35 | 43 | 40 | 35 |
| | $55 \times 55$ | **51** | 59 | 52 | 63 | 60 | 54 | **52** | 55 | 65 | 61 | 58 |
| | $60 \times 60$ | **72** | 78 | 75 | 83 | 78 | 73 | **71** | 77 | 89 | 79 | 84 |
| Sliding puzzle | $5 \times 5$ | **1521** | 1558 | 1534 | 1559 | 1545 | **1524** | 1539 | 1533 | 1556 | 1544 | 1531 |
| | $6 \times 6$ | **2322** | 2353 | 2334 | 2439 | 2388 | 2321 | **2326** | 2329 | 2334 | 2329 | 2329 |
| | $7 \times 7$ | **3343** | 3375 | 3347 | 3431 | 3411 | 3421 | **3356** | 3449 | 3512 | 3448 | 3379 |

Table 6: Average number of expanded states.

| problem | complx. | SBA* | FDSS | A* | | | | | GBFS | | | | | |
|---|---|---|---|---|---|---|---|---|---|---|---|---|---|---|
| | | | | $L^*$ | $L_{gbfs}$ | $L_{rt}$ | $L_2$ | $L_{be}$ | $L^*$ | $L_{gbfs}$ | $L_{rt}$ | $L_2$ | $L_{be}$ | $L_{le}$ |
| blocks | | 100 | 100 | 100±0 | 100±1 | 100±0 | 99±1 | 100±0 | 100±0 | 100±0 | 100±0 | 100±0 | 100±0 | 99±1 |
| ferry | | 100 | 95 | 98±3 | 98±3 | 100±0 | 92±8 | 100±0 | 98±3 | 100±0 | 100±0 | 100±0 | 98±4 | 98±3 |
| npuzzle | | 100 | 89 | 89±1 | 87±2 | 88±0 | 83±4 | 89±1 | 92±5 | 89±1 | 89±1 | 89±1 | 92±7 | 88±3 |
| spanner | | 100 | 90 | 100±0 | 89±2 | 100±0 | 84±6 | 92±6 | 100±0 | 100±0 | 100±0 | 100±0 | 100±0 | 100±0 |
| elevators | | 100 | 37 | 91±2 | 85±10 | 75±8 | 36±6 | 66±3 | 92±3 | 85±11 | 79±8 | 76±3 | 67±4 | 58±11 |
| Sokoban | 3 boxes | 100 | 100 | 99±0 | 98±0 | 96±0 | 97±0 | 92±0 | 98±0 | 100±0 | 94±0 | 95±0 | 92±0 | 98±0 |
| | 4 boxes | 100 | 81 | 89±2 | 89±1 | 85±2 | 81±0 | 82±3 | 87±2 | 91±1 | 84±4 | 83±3 | 84±3 | 84±3 |
| | 5 boxes | 97 | 67 | 80±1 | 75±2 | 72±2 | 72±1 | 73±0 | 78±1 | 77±1 | 74±3 | 72±3 | 72±2 | 73±1 |
| | 6 boxes | 55 | 49 | 76±2 | 69±1 | 59±3 | 51±0 | 53±3 | 73±1 | 71±1 | 56±0 | 512 | 54±0 | 64±1 |
| | 7 boxes | 46 | 31 | 55±4 | 49±3 | 47±4 | 42±3 | 45±5 | 51±3 | 49±3 | 48±3 | 43±2 | 45±2 | 49±3 |
| Maze w. t. | 50 × 50 | 92 | 75 | 92±0 | 91±0 | 88±1 | 87±1 | 87±0 | 89±0 | 90±1 | 89±1 | 84±0 | 85±1 | 89±0 |
| | 55 × 55 | 52 | 50 | 78±1 | 75±4 | 73±1 | 72±3 | 74±3 | 74±0 | 75±3 | 74±2 | 72±3 | 75±3 | 74±2 |
| | 60 × 60 | 0 | 0 | 49±1 | 37±0 | 35±3 | 32±2 | 31±1 | 42±3 | 48±1 | 36±3 | 34±3 | 32±2 | 42±0 |
| Sliding puzzle | 5 × 5 | - | 1 | 88±1 | 83±3 | 84±3 | 80±2 | 82±0 | 86±3 | 87±1 | 84±0 | 84±1 | 84±1 | 85±1 |
| | 6 × 6 | - | - | 51±2 | 48±3 | 49±2 | 45±4 | 46±3 | 47±3 | 49±3 | 45±2 | 43±3 | 66±2 | 48±3 |
| | 7 × 7 | - | - | 39±3 | 35±3 | 36±3 | 32±3 | 34±3 | 35±3 | 36±3 | 35±3 | 32±3 | 34±3 | 35±3 |

Table 7: Average fraction of solved problem instances in percent with standard deviation. SBA* and FDSS denotes Fast Downward Stone Soup, They are domain independent planners.

| | | A* | | | | | GBFS | | | | | |
|---|---|---|---|---|---|---|---|---|---|---|---|---|
| problem | complx. | $L^*$ | $L_{gbfs}$ | $L_{rt}$ | $L_2$ | $L_{be}$ | $L^*$ | $L_{gbfs}$ | $L_{rt}$ | $L_2$ | $L_{be}$ | $L_{le}$ |
| blocks | | 21.6 | 22.7 | **21.4** | 22.5 | 22.9 | **21.8** | 22.7 | 22.1 | 22.5 | 23.1 | 34.1 |
| ferry | | 16.8 | 16.9 | **16.7** | 16.8 | 16.8 | **16.8** | 16.9 | **16.8** | **16.8** | **16.8** | 16.9 |
| npuzzle | | 19.2 | 23.5 | 18.2 | 17.5 | **17.3** | 37.6 | 36.4 | 39.6 | 35.1 | **34.1** | 124.6 |
| spanner | | **49.1** | 49.2 | 49.2 | 49.2 | **49.1** | **49.0** | 49.2 | 49.1 | 49.2 | 49.3 | 49.1 |
| elevators | | 16.1 | 16.4 | 14.7 | 17.9 | **15.8** | 19.4 | 16.6 | **15.9** | 18.0 | 16.1 | 21.4 |
| Sokoban | 3 boxes | **13.1** | 13.3 | 13.8 | 13.2 | 13.4 | **13.1** | 13.2 | 13.2 | 13.5 | 13.2 | 13.2 |
| | 4 boxes | 15.4 | **15.2** | 16.1 | 15.7 | 16.8 | 16.7 | **15.1** | 16.1 | 15.6 | 16.3 | 15.8 |
| | 5 boxes | 20.1 | **19.2** | 21.3 | 22.1 | 20.9 | **19.8** | 20.4 | 21.3 | 22.1 | 19.9 | 20.1 |
| | 6 boxes | 29.6 | 29.3 | 27.7 | 28.2 | **26.8** | 28.3 | **28.1** | 29.6 | 29.4 | 29.9 | 30.1 |
| | 7 boxes | 31.9 | **31.4** | 35.4 | 33.1 | 34.1 | **30.1** | 33.3 | 35.2 | 32.7 | 34.0 | 35.5 |
| Maze w. t. | $50 \times 50$ | **24.1** | 25.3 | 25.1 | 24.3 | 24.3 | **24.3** | 24.5 | 25.4 | **24.3** | 25.4 | 24.7 |
| | $55 \times 55$ | 34.1 | **33.2** | 35.0 | 34.2 | 33.9 | **33.2** | 33.1 | 34.6 | 34.6 | 36.5 | 36.3 |
| | $60 \times 60$ | **41.2** | 42.9 | 41.4 | 43.2 | 42.8 | **42.1** | 43.6 | 44.2 | 45.2 | 45.3 | 45.1 |
| Sliding puzzle | $5 \times 5$ | **150.1** | 153.7 | 155.2 | 154.6 | 154.5 | 154.5 | 153.5 | 153.9 | 155.0 | **151.1** | 152.1 |
| | $6 \times 6$ | **252.3** | 254.2 | 253.8 | 254.8 | 254.0 | **255.9** | 256.4 | 255.3 | 256.2 | 254.9 | 256.3 |
| | $7 \times 7$ | **321.1** | 324.1 | 322.2 | 324.3 | 320.4 | **322.9** | 324.1 | 323.4 | 327.1 | 324.6 | 323.7 |

Table 8: Average number of length of the solution. The average is computed only over problem instances solved by all 11 variants of forward search.

