# OpenReview forum: "Optimize Planning Heuristics to Rank, not to Estimate Cost-to-Goal"
_NeurIPS.cc/2023/Conference — NeurIPS 2023 poster_

### Official Review · Reviewer_ahr3 · 2023-06-26

**Soundness:** 2 fair
**Presentation:** 3 good
**Contribution:** 3 good
**Rating:** 6
**Confidence:** 4

**Summary:**

The paper highlights the limitations of the conventional method for learning heuristics in search algorithms, which involves imitating the optimal cost-to-go function. The authors present an alternative approach that focuses on optimizing the correct ranking of state pairs by the heuristic. Because of that characteristic, the authors argue that the resulting heuristic offers both theoretical and practical advantages compared to the standard approach.

**Strengths:**

The paper presents an original idea that indeed can offer a significan advantage for planning methods. The presentation is mostly clear, with only minor problems.

**Weaknesses:**

In my opinion, the claims formulated in the paper should be supported better. I'm not convinced that the reported results stem from the theory. Clearer discussion of theoretical advantages and disadvantages of various approaches, more detailed experiments including analysis of the learned heuristic, and comparison with some other baselines could be a significant improvement to the paper. Details of my concerns are discussed in the comments.

**Questions:**

In general, I like the idea and the paper, although at this point it may need some improvements. At this point I'm not fully convinced that the improvement you show is due to the theoretical advantages (which should be clarified) or just a fancy heuristic that happens to work better than a certain imitation objective. Below I highlight a few specific comments.

- l.111,117,157 Arguably a central expression, appears in 3 places, and every time is written differently. Correct it (or explain that it's actually not the same).

- l.113-125 This explanation of the notation above is a lifesaver.

- l.131 Does there exist at least one perfect ranging for every problem? Semms that h(s):=[-infty on a fixed optimal path, infty otherwise] works. What happens in quite a common case in which s_0 is not specified and the heuristic should work for any starting state? Does there always exist a perfect ranking in such case?

- l.105-150 While I don't deny the soundness of the theory, it reads a little like explaining quite an intuitive idea ("the heuristic is strictly optimal iff in every step of the search the next node on a fixed optimal path receives the lowest value") with very complicated expressions. I'd consider changing the order and starting from intuition, moving to the formalities afterwards. The ultimate conclusion should be the expression in l.111, which I believe is the central object in this section.

- l.161 Quite a strong assumption.

- l.167 The relation between log(1+exp(x)) and [[x>0]] indeed seems loose. On convergence it will have the desired property indeed, but it doesn't exactly optimize the number of violations, rather a sum of violation volumes. How does it relate to your theory? An experimental analysis of the learned function would do.

- l.180 I'd consider rewording the section title. Although after reading the section it makes sense, at first it seems like a general discussion of optimizing neural networks.

- l.181-185 I like the idea, but it's not true that it is easier to learn a function from a bigger class. The core element is the learning signal. In case of cost-to-go, where we have actually a direct supervision, it is of much better quality than in case of a set of inequalities. Consider an example in which we have n states and we want to learn any monotonic function. Although there are much more functions satisfying f(s_{i+1})>f(s_i), learning fixed supervised targets should be faster.

- l.186-196 "states off the solution path" -- you mean in the training dataset or in general? As far as I understand, your training dataset consists of solving paths only, so the issue applies to your ranking loss as well, doesn't it? When provided with off-the-path states, standard cost-to-go functions can also utilize them for learning (e.g. bootstrapping for Q-learning). The issue you describe is devoted to the particular training loss you consider, not learning cost-to-go in general.

- l.203-207 I don't understand. Optimizing the proposed losses never considers dead-ends as they don't lie on the optimal paths. Do you consider any hand-modified heuristics? If you are worried about too high gradients, use Huber loss instead of L2. Why should lare values appear in the training? (5) is strictly supervised.

- l.269 That's a shame, it would support your claims well as it makes a middle-ground between (3) and (5).

- l.285 How costly is it to get this dataset? Is your method more efficient than SymBA* afterwards? Gathering optimal paths for training may be very costly, especially in the hardest domains. Thus, can you comment on the applicability of your claims, since you require a costly operation of gathering optimal paths for your training> Can your method, like standard imitation, learn also from sub-optimal data, which is often much easier to get?

- l.286 Why a single problem instance?

- l.322-342 If the test sets are harder than training sets, how does it relate to the theory in which you assume having optimal paths?

- l.343-363 I'm not familiar with the complexity of some instances of the environments. Thus, it would be helpful to see some trivial baselines in the comparison, such as BFS or Dijkstra. Also, to showcase an advantage of your theory-driven approach, I'd consider comparing with a simple objective that learns to output 0 on the solving paths and 1 on other states outside the path. It fits the definition of optimality and doesn't seem to be stable or generalizable (or maybe?). Advantage over such a baseline would highlight the benefit of learning the ranking over heuristics. It's missing now. Yes, I know that you compare already agains many relevant baselines.

- l.343-363 Also, the claims should be supported by some ablations. Is the learned heuristic correct? Does it learn to properly rank states, even in the training domains? What does it converge to, do the values resemble cost-to-go or some other interpretable quantities? What are the mistakes? Does the learned heuristic fit the theory of what it should learn? Are the paths obtained this way optimal? How do they relate to optimal paths in terms of length? How often only states on an optimal path are expanded? Can you learn a useful heuristic for a fixed domain with many possible starting points?

- l.375 I'm not sure whether optimizing set of inequalities is easier than a direct supervision.

And a few minors:

- l.31,212,215,271 etc. When you refer to something, either in your paper or in supplementary, refer to the exact section.

- l.40 Do you mean "state-of-the-art"?

- l.48 I wouldn't say "always", you can demonstrate that it's better "in those eight environments".

- l.50-72 I can't find a definition of g(s) or h(s) before you use it to explain your algorithm. Maybe in preliminaries? g(s) is defined below the algorithm, but h(s) is not.

- l.116,135, and others: I feel confused whether "heuristic" refers to h or f. Please disambiguate the notation.

- l.305 24000 gradient steps in 100 GPU hours? Seems quite slow, I'd suggest checking it.

**Limitations:**

The limitations are not clearly stated, there should be a dedicated section for them. Societal impact is not an issue here.

---

> ### Author Rebuttal · Authors · 2023-08-07
>
> We thank the reviewer for valuable commplents. Due to space limit of, we will address his comments in the same order using lines for perfect matching.
> * *l.111,117,157* We seen the expression in lines L.111 and L.117 the same. Can you clarify us, where do you see the difference exactly? The expression on Line L.157 differs in not using and index $j$ for identifying the state, because it was not needed.  We will happily reintroduce state $s_j$ back, if it helps readability.
>
> * *l.113-125* We agree.
>
> * *l.131* On line L.144 we state that the complexity of perfect ranking heuristic is equal to the complexity of solving the problem, because we can solve the problem and then create the heuristic by your construction. Since the initial state $s_0$ is part of the definition of the problem instance, to assert that heuristic is perfect ranking for any initial state, we would need to verify the conditions for all initial states.
>
> * *l.105-150* Changing the order sounds like a good idea.
>
> * *l.161* This assumption seems to be common in papers about imitation learning (e.g. Ref. [9,17,19,20,45,49] in the paper). We prefer this over bootstrapped learning (arguably more realistic) to better expose differences in loss functions while isolating other sources of variability.
>
> * *l.167* We do not see a problem in using this relation, since it is used in majority machine learning methods optimizing parameters by gradient descend.  The relation is loose, since  log(1+exp(x)) upper bounds [[x>0]], but it allows methods scale. We can optimize parameters of NN [[x>0]] using mixed integer program, but this does not scale to interesting problems. The theory is sound  [[x>0]], but log(1+exp(x)) is needed to make it practical.
>
> * *l.180* This is an interesting view, but we prefer to keep it as is, because the proposed loss function can be used with other machine learning methods like support vector machines or gradient-boosted decision trees. One of the motivation for our work stems from the fact a lot of attention is paid to architecture of NNs (or classifiers in general), but loss functions are neglected despite being very important.
>
> * *l.181-185* "Easy to learn" is an unfortunate  expression. A better would be that more functions can satisfy the conditions. The quality of cost-to-go is debatable (and we would love to debate it more in discussion). As argued in the paper, search does not care about values, but about their order. For cost-to-go work, you also need to learn its values for states off the optimal path, otherwise it fails as discussed in the paper. A nice example, where cost-to-go gives wrong signal is when all actions have zero cost (but the proposed method still works).
>
> * *l.186-196* We consider L2 loss as is used in imitation learning in planning, which uses exclusively states on the optimal path, because otherwise the cost-to-go is not known. If it uses states off the optimal path, it needs to run solver to get the correct values, which corresponds to solving a new problem. Repeated simulations in Q-learning corresponds to repeatedly running the solver, which is different settings.
>
> * *l.203-207* If you use just cost-to-go on optimal path, in Equation (5), you will never learn about dead-ends. But as [9] has shown, informing NN about dead-ends by giving them high heuristic value in the training set helps. We tried to explain that dead-ends are considered by the proposed loss function without any modification, since they will be among states off the optimal path and heuristic is learnt to avoid those.
>
> * *l.269* We have some initial results for comparison and we can share them with you in discussion phase. The word limit is quite tight.
>
> * *l.285* Obtaining a training dataset is costly, but this step is required by methods optimizing cost-to-go by $L_2$ loss as well. Therefore we do not see this as a problem of the proposed method. We emphasize that experiments on grid domains show that training on simple problems (e.g. Sokoban with 3 boxes) allow to generalize to hard ones and better than learning cost-to-go (GBFS with $L_2$ can solve 42% problems with 7 boxes while A* with L* can solve 55%.
>
> * *l.286* Because all states in minibatch are used in all known inequalities.
>
> * *l.322-342* The relation is through generalization bounds of a learnt classifier. The argument is the same as for learning cost-to-goal by minimizing $L_2$ loss.
>
> * *l.343-363*  We did not did this, because we have not seen it in the prior art (Refs. [9,17,19,20,37,39,45,49,53]), from where we have taken relational problems. The fraction of solved problems for BFS (5s time limit as used by solvers in the paper) is provided below.
> ```
> problem       fraction
> ────────────────────
> blocks        0.35
> ferry         0.31
> npuzzle       0.14
> spanner       0.63
> elevators     0.22
> ```
> Your example of learning zero on optimal path and one off the optimal path is a realization of the proposed method with regression ($L_2$) loss. It is an interesting experiment.
>
> * *l.343-363* These are allinteresting questions, but it will not fit the page limit. The rebuttal for the reviewer jMYD contains a maze without obstacles, where all solution path are of the same costs. This example answers some of your questions.
>
> * *l.375* When you optimize inequalities, you do not have to be exact and you are not penalized when the estimate of cost-to-goal is perfect. We think that also for humans, it is easier to estimate that something is closer / farther than how far it is exactly (assuming the human is not equipped with smartphone and maps).
>
> * The computation complexity is reported for all experiments including hyper-parameter search and development in case of relational domains. The training time for a single grid domain is under 2 GPU/h, the training time for single relational domain is usually around 1/2 CPU/h except gripper, where it takes about 8 CPU/h (see rebuttal to reviewer 458p).

---

> > ### Comment · Reviewer_ahr3 · 2023-08-17
> > **Thank you for the response**
> >
> > - l.111,117,157 In l.117 there seems to be an extra conjunction, in l. 157 $s_j$ is changed to $s$ and the superscript of $\mathcal S$ seem to be missing $\pi$. While these are all details, they make the expressions less readable.
> > - l.131 I mean that Theorem 1 provides an equivalent condition for a perfect ranking, but it is also important to know whether the definition is robust enough that a perfect ranking exists at all. As you said, we know it if the initial state is specified. I'm curious about a case when the initial state is not specified. Is it true that for every environment there exists a function that is a perfect ranking for every possible initial state? (it would be even better to understand how does it look like) Can you give an intuition why it holds, or a counterexample? Proving such a theorem would make your paper stronger. The trivial case of "1 on an optimal path, 0 elsewhere" is clearly too strong for the general case. Is your definition robust enough to cover all possible scenarios? P.S. I think no: if we have four states A,B,C,D, the goal state is A, and d(A,B)=d(A,C)=d(B,D)=1 and d(A,D)=d(B,C)=9 there is no ranking that is perfect for both C and D since they both expand to {A,B} but require different ordering. However, if you include the information about the last action executed, such a ranking always exists (the graph of optimal actions is well-defined, so you can estimate 0 there and 1 for other actions). While such an analysis may be out of the scope of this paper, I suggest thinking about it.
> > - l.186-196 Can you also clarify whether your method suffers from this issue? I understand that you somehow use states off the solving path during training. Where do they actually come from? Could you refer to the place in the paper where it is explained, I can't find it now.
> > - l.343-363 I acknowledge the results you provided. I think that such a discussion strengthens the presented results, I suggest including it in the paper, even in an appendix. I see that even in such a small example, the magnitude of the learned estimates is considerable. Do you somehow control it to ensure no blow-up?

---

> > > ### Author Response · Authors · 2023-08-17
> > > **Is this a solution to your example?**
> > >
> > > * Regarding the l.117. and l.157, we think we understand your point now and will easily fix it. We have not thought the differences between $s_j$ and $\mathcal{S}$ to be disturbing, but we are happy to know about them now and change it according to your suggestions to improve the readability.
> > >
> > > * Now, we understand your point and we think the function should exist if we do not take into the account its computational complexity (which is a helper to use the solver to identify optimal solution path). But we need to think about the formal proof and it would be probably a different work. Especially we need to think what can go wrong.
> > >
> > > We have tried to solve your example on the paper and we think the solution exist. we will talk exclusively about A*, because GBFS does not have a guarantee on optimal and therefore the quality of the solution does not matter.
> > >
> > > Imagine the heuristic values of states to be h(D) = h(C) = h(A) = 0, and h(B) = 4.
> > > Let's now simulate the behavior of A* for different starting states {B,C,D}
> > >
> > > **Starting at D:** A* will put to open list states A and B. The merit function $f$ will be: $f(A) = 9 + h(A) = 9$ and $f(B) = 1 + h(B) = 5.$ A* takes from open list state B, check if it is a goal (it is not) and expand it to A. When inserting A to open list, A* founds a better bath to A and updates the path to A to ((D,B), (B,A)). The open list contains only one state A, which is the solution state and optimal solution was found.
> > >
> > > **Starting at C:** A* will put to open list states A and B. The merit function $f$ will be: $f(A) = 1 + h(A) = 1$ and $f(B) = 9 + h(B) = 14.$ A* takes from open list state A, check if it is a goal (it is), and return solution path (C,A).
> > >
> > > **Starting at B:** A* will put to open list states A and C. The merit function $f$ will be: $f(A) = 1 + h(A) = 1$ and $f(C) = 9 + h(C) = 10.$ A* takes from open list state A, check if it is a goal (it is), and return solution path (B,A).
> > >
> > > Unless we understand incorrectly your example, the optimal ranking $h$ exists.
> > >
> > >
> > > * l.186-196 We collect states off the optimal path by expanding states on the optimal path (of a given problem instance). In our implementation, we do this when we are preparing mini-batches, as the complexity is trivial. We might not have discussed this in sufficient detail, since we considered this to be an implementation issue. We have believed the need to collect these states to be clear from the theoretical exposition of the loss. We now see that adding a sentence about this to the experimental settings might make it more clear.
> > >
> > > * l.343-363  The large values of heuristic values in the example were caused by the logistic loss convex surrogate, which is nonzero and strongly convex. A hinge loss would create a different value which might be way smaller. As we have mentioned on the L.182, the solution set is invariant to strictly monotone transformations (in case of A* algorithm, the set of transformation is smaller),  therefore the optimization algorithm has picked a solution and did not care about size of heuristic values.

---

> > > > ### Comment · Reviewer_ahr3 · 2023-08-19
> > > >
> > > > > We will talk exclusively about A*, because GBFS does not have a guarantee on optimal and therefore the quality of the solution does not matter.
> > > >
> > > > As far as I understand, your theory covers all combinations of $\alpha,\beta$, which include GBFS and nearly-GBFS planners. In fact, when it comes to theory, such instances are more interesting since, as you observed in Example 1, for A* the existence of perfect ranking is easy -- it is the cost-to-go function (up to the tie-breaking issue).
> > > > Thus, also in my example, the cost-to-go is a perfect ranking for A*. If $\alpha,\beta > 0$, we can always rescale $h(s)$ to pretend that $\alpha=\beta$. However, this is not the case with GBFS ($\alpha=0$). In my example, I think there is no optimal ranking for GBFS, because of the reasons I provided.

---

> > > > > ### Author Response · Authors · 2023-08-19
> > > > >
> > > > > We think that for GBFS the optimal ranking exist as well. For the case you have provided it will be as follows: h(d) = 2, h(b) = h(c) = 1, h(a) = 0, which it is equal to the number of nodes that needs to be expanded to reach the goal.
> > > > >
> > > > > Similarly as above, we can verify that GBFS is optimally efficient as it will find the solution while expanding the minimal number of nodes.
> > > > >
> > > > > **Starting at D**: GBFS will put to open list states A and B with heuristic values h(A) = 0 and h(B) = 1. GBFS takes A from the open list and checks if it is a goal state. Since A is goal state, GBFS terminates returning path (D,A) as a solution.
> > > > >
> > > > > **Starting at C**: GBFS will put to open list states A and B with heuristic values h(A) = 0 and h(B) = 1. GBFS takes A from the open list and checks if it is a goal state. Since A is goal state, GBFS terminates returning path (C,A) as a solution.
> > > > >
> > > > > **Starting at B**: GBFS will put to open list states A and C with heuristic values h(A) = 0 and h(C) = 1. GBFS takes A from the open list and checks if it is a goal state. Since A is goal state, GBFS terminates returning path (B,A) as a solution.
> > > > >
> > > > > The optimal ranking for all states for GBFS therefore exist, as it always finds the solution while expanding minimal number of states. But the ranking function does not guarantee that the GBFS will find optimal solution. If we are not mistaken, we have not claimed in the paper that GBFS with optimal ranking function is guaranteed to find optimal solution. We claimed it will find the solution while expanding minimal number of states. We also said (with proof in suplementary) that if all actions have same cost, then optimal ranking for A* is optimal ranking for GBFS, but this is not the case of this problem.
> > > > >
> > > > > This example nicely shows that optimal ranking is **search dependent**. We will try to explain this difference more explicitly and add this example into the supplementary, if you agree.

---

> > > > > > ### Author Response · Authors · 2023-08-19
> > > > > > **I got it**
> > > > > >
> > > > > > We now understand your point and your are right. For GBFS, there is no optimal ranking for all states. This is because the definition of *optimal ranking* requires the inequalities to hold for the optimal path. Thus you are right that the heuristic does have to exist for all initial states and your example is correct.
> > > > > >
> > > > > > The problem can be fixed if the definition of optimal ranking is changed to consider two cases: $\alpha > 0$ and $\beta > 0$ and $\alpha = 0$ and $\beta > 0$. In the first case, the optimal ranking should be defined with respect to the "optimal path" (this is the A*), in the latter  it should be path with minimal number of expansions" for GBFS, because with GBFS the user wants to found a solution, but should care about its optimality. With this change, the heuristic function will exist for the above example.
> > > > > >
> > > > > > Thank you very much for this discussion. It was very enlightening and we appreciate your input.

---

> > > > > > > ### Comment · Reviewer_ahr3 · 2023-08-20
> > > > > > >
> > > > > > > Thank you for the discussion and clarifications. I acknowledge that by increasing my score.

---

> > > > > > > > ### Author Response · Authors · 2023-08-21
> > > > > > > >
> > > > > > > > Thank you very much. We appreciate the discussion.

---

### Official Review · Reviewer_458p · 2023-07-02

**Soundness:** 3 good
**Presentation:** 4 excellent
**Contribution:** 3 good
**Rating:** 6
**Confidence:** 4

**Summary:**

Planning with (possibly learned) heuristic function is important for both planning and learning communities. The paper focuses on the forward search without expanding redundant search states in their forward process. With the development theorem, the authors demonstrated how the proposed concept work in various domains with A* and GBFS algorithms.

**Strengths:**

- The concept of the focusing problem, wheares they seem to be simple, is interesting. The proposed theoretical discussion (e.g., sufficiency and necessity in Theorem 1) is clearly explained; the whole paper is well written and structured.
- The proposed approach with $L_\star$ based on the two fundamental search techniques A* and GBFS work efficiently than counterparts.

**Weaknesses:**

- Although the proposed learning-based approach is promising, it requires much learning time on PDDL domains (e.g., 17724 CPU hours).
- The scales and variations of compared benchmark domains seem limited (e.g., traditional search problems and mazes).

**Questions:**

The paper is very clearly organized, and I can focus on the content. The below items are come comments.

- Learning-based heuristic functions could be promising for their generalization ability. However, sometimes they are argued that the required learning times (or resources) are too large. In the reported times with PDDL domains, it requires 17724 CPU hours. These results may cause some questions about the efficiency of the learning-based heuristic itself.
    - Do you discuss why the learning-based heuristic is promising, even though the training resource is required?
    - Can we compare planners without learning and planners with learning (i.e., the proposed method) in terms of total required resources?
    - Can the two solvers (A* and GBFS) have similar characteristics in terms of learning-based methods? The two solvers can be formulated similarly, as noted in L.93. Can we regard these methods as just variants of search using the function of the form in L.92, or can we have some important findings of these methods through the experiments?
- Mathematical notation issues:
    - In Def. 1, some mathematical errors seem to occur in the second condition: see L.111 and L.117 to check the condition of $\wedge$.
    - In Eq.(6), the log() is not closed.

**Limitations:**

I have no explicit concerns about the limitations.

---

> ### Author Rebuttal · Authors · 2023-08-05
>
> Thank you for an interest in our work.
>
> *Do you discuss why the learning-based heuristic is promising, even though the training resource is required?*
> Let us first clarify the computational complexity. The paper reports the total time of all experiments presented in the paper. This means that 17724 CPU hours is cost of all experiments (training, evaluation, search for hyper-parameters) on five relational domains, six loss functions, and for three repetitions. Moreover, since we have assumed that Neurips welcome reporting the computation time needed for development, these cost include the development (we took it from an audit of SLURM scheduler during period of authors working on this paper). Similarly, 100 GPU hours and 1000 CPU hours is an upper bound (we did not have the logs from the scheduler due to different computation environment) of the training time (GPU cost) and evaluation time (CPU costs) of three repetitions of three grid problems for six loss functions.  **The time to train a neural network took for a single problem and single loss functions** are
> 1. approximately 1.85 GPU/h for grid domains;
> 2. and usually 0.5 CPU/h for relational domain. The exception is Gripper domain, where the training takes about 8h due to large branch factors. We believe this training time can be reduced by different construction of minibatches, but we consider this to be outside of this paper.
>
> From these number we believe the training times are acceptable even for practical use. We emphasize though, that the training time does not include the cost of creating the training set. We will improve the formulation in the main paper, since it seems to be confusing.
>
> *Can we compare planners without learning and planners with learning (i.e., the proposed method) in terms of total required resources?* This might be difficult, because we do not know the development cost of classical planners. Specifically, we do not know, how much CPU time was used for their tuning and testing. The classical planners and those with heuristics realized by NN were executed on the *same* hardware (always only single core CPU) and were given the same time limit (10mins for grid domains, 5 seconds for relational domains), which should make it fair comparison.
>
> *Can the two solvers (A* and GBFS) have similar characteristics in terms of learning-based methods? The two solvers can be formulated similarly, as noted in L.93. Can we have some important findings of these methods through the experiments?* We also believe that the proposed loss functions, the methods are related. The experimental results, where we have observed that NN optimized for A* works well with GBFS (but not the other way around) motivated the formal proof of Statement on line L.350 (the proof is in the appendix).
>
> *Mathematical notation issues*
> We have checked lines 111 and 117 and believe them to be correct (albeit L.111 contains additional $\wedge$). $s_k \in \mathcal{S}^{\pi_{:,i-1}}$ says that $s_k$ has to be on a solution path between start and state $s_i$ and $s_j \notin \mathcal{S}^{\pi_{:,i}}$ should be outside of the solution path. Thus $s_j$ is distance one from the solution path. We completely agree with other reviewers that this is part is difficult to read and we will try to improve it.
>
> Than you very much for spotting the missing parentheses in Equation (6).

---

> > ### Comment · Reviewer_458p · 2023-08-17
> > **Thank you for clarifying the concerns, particularly for total costs.**
> >
> > I appreciated the updates from the authors.
> >
> > I may misunderstand something about the learning points. However,  the responses above clarified some of them, and I understand some argued difficulty in considering the total development costs (for classical planners).
> >
> > For the findings (of A*-based and GBFS-based learning planners) in the paper, I agree that the findings are interesting for the literature. As I'm not so frustrated with the notations, but it seems to be better for improving the readability (as other reviewers commented). Any way, thank you for your comments and contributions.

---

### Official Review · Reviewer_7xum · 2023-07-04

**Soundness:** 3 good
**Presentation:** 3 good
**Contribution:** 3 good
**Rating:** 5
**Confidence:** 3

**Summary:**

This paper investigate the computational advantage if a forward-search algorithm utilises ranking rather than optimising cost-to-goal. I think the contribution is good in general.

**Strengths:**

1. The problem is formulated well.
2. The introduction of the contents is suitable.
3. The experiment section is good.

**Weaknesses:**

1. The definition of "optimal ranking" is so important to this paper that it needs to be presented before first used. The authors should not use "(definition is below)" for this term.
2. I suggest the authors simplify some expressions. For example, at some points where strict rigorousness is not required, "minimise the number of expanded states"--> "efficient".
3. I cannot understand line 7 in the supplementary file. Is it grammatically correct?
4.Line 10 in the supplementary file, it is not "lastly". By the way, what is the connection between the Section 2 in the supplementary and the main text? I don't think there is a need to present this section. As long as "estimating the cost-to-go converges" is slower than the proposed scheme, the contribution of the main text is valid. Am I correct?
5.Can the author find a citation for the claim of Example 1? so that there is no need to present it in detail. I think this example broke the reading flow.
6. Why is the related work section at Sec 5? Why can't it be Sec 2?
7. I don't think Sec 4.3 presented rigorous statements. All things are subjective. Did the author write "conclusive remarks" here? I suggest the authors move them to the conclusion section. Don't be afraid that the algorithm section is too short.

Minor points:

1. I didn't find the definition for the \wedge symbol in Def.1.

**Questions:**

Please explain my comments listed in the weakness block.

**Limitations:**

I don't think the paper has any negative societal impact. But I do suggest the authors polish their Sec 4.3.

---

> ### Author Rebuttal · Authors · 2023-08-06
>
> *The definition of "optimal ranking" is so important to this paper that it needs to be presented before first used. The authors should not use "(definition is below)" for this term.*
>
> This is a good suggestion. We will incorporate the definition to the motivation.
>
> *I suggest the authors simplify some expressions. For example, at some points where strict rigorousness is not required, "minimise the number of expanded states"--> "efficient".*
>
> We are totally aware that some wordings are clumsy, but we wanted to be rigorous. Specifically, the planning community consider A* with true cost-to-goal heuristic to be "efficient", but it is not without additional tie-breaking rules. We were afraid that if we would not be precise, some readers might be confused.
>
> *I cannot understand line 7 in the supplementary file. Is it grammatically correct?*
> There seems to be some typos. We wanted to say that making the learnt heuristic function $h$ to be goal-aware can be difficult, when the problem instance contains multiple goals not defined by a single substate.  All heuristic functions defined by neural networks known to use have this problem and we consider it outside of the scope of this paper. Since A* and GBFS planning algorithms do not need the heuristic functions to be goal aware, we do not see this feature to be an issue, but we wanted to be clear.
>
> *Line 10 in the supplementary file, it is not "lastly".*
> Yes, we will delete this
>
> *By the way, what is the connection between the Section 2 in the supplementary and the main text? I don't think there is a need to present this section. As long as "estimating the cost-to-go converges" is slower than the proposed scheme, the contribution of the main text is valid. Am I correct?*
> We actually believe this to be important, as it is another argument supporting to learn the heuristic function by "ranking" instead of by learning to estimate cost-to-goal. In planning by imitation learning, creating the training set is usually done by running a solver, which is expensive. This argument therefore use the statistical learning theory to argue that learning to rank is simpler problem than regression, because it converges faster with respect to the number of samples. This makes it cheaper. Since we have seen this claim in papers about ranking (not in the planning domain) without a proper theoretical justification, we wanted to have a theoretical evidence with references. Section 2 in supplementary therefore fill this gap by composing results of two papers [8] and [54] (citation numbers of the paper), such that the convergence rates can be compared side-by-side. Moreover, this helps to highlight a difference between sample in regression (one state) and sample in ranking (pair of states). Ref. [8] seems to be relatively unknown, judging by fact it has "just" 228 citations while being 24 years old.
>
> *Can the author find a citation for the claim of Example 1? so that there is no need to present it in detail. I think this example broke the reading flow.*
> The example is adopted from Ref. [14] (citation numbers of the paper). We add the example to make the paper more self-contained. If you think it breaks the flow, we would happily remove it.
>
> *Why is the related work section at Sec 5? Why can't it be Sec 2?*
> The position of related work is questionable. We had the impression that NEURIPS community typically puts the related to work towards the end of the paper (e.g. [A] below), because the reader is usually eager to read the novel content of the paper. On the other hand moving the related work to Sec. 2 introduces the reader better to the problem. We do not have a strong opinion and would be happy to move it to Sec. 2.
>
> *I don't think Sec 4.3 presented rigorous statements. All things are subjective. Did the author write "conclusive remarks" here? I suggest the authors move them to the conclusion section. Don't be afraid that the algorithm section is too short.*
>
> We very kindly disagree with the reviewer and would like to ask him to identify, which statements are subjective and why. There are following statements:
> 1. Size of the solution set
> 2. Optimizing cost-to-goal does not utilize states off the solution path
> 3. Cost-to-goal provides a false sense of optimality
> 4. Heuristic value for dead-end states
> 5. Goal-awareness
> 6. Speed of convergence against the size of training set
> 7. Conflicts in training set
>
> Statements 2,3,4 and 6 are supported by the theory in the paper. Statement 1 might be considered speculation, but it is very likely true (in the paper we have written that we lack the formal proof). Statement 5 is in favor of estimating cost-to-goal, since the proposed heuristic can be made goal-aware only in some cases. Finally, Statement 7 is a statement. We agree this can be moved to discussion.
>
>
> [A] Klissarov, Martin, et al. "Adaptive Interest for Emphatic Reinforcement Learning." Advances in Neural Information Processing Systems 35 (2022): 95-108.

---

> > ### Comment · Reviewer_7xum · 2023-08-17
> >
> > Thank you for the rebuttal. Sorry for my delayed response. Please see my response below.
> >
> > 1. Rigorous issue: I don't think the authors understood me. The authors should know that when strict rigorousness cannot be established by non-clumsy wordings (here it is the case), then it is better to present the thing such that what the reader intuitively thinks is exactly what you want to present. As a counterexample, there are many "efficient" in the paper, but the authors should not expand all of them to "minimise ...".
> >
> > 2. Description in supplementary issue: I don't think the authors need to say things like line 7. (In my opinion supplementary did not do a good job actually.)
> >
> > 3. I agree with the authors' opinion that "ranking" is probably easier to estimate than "cost-to-goal". I did search related works for a while (therefore I'm late for the response, apologise again for the delay) and found one regarding this:
> >
> > [1] Yang T, Huang L, Wang Y, et al. Efficient search of the k shortest non-homotopic paths by eliminating non-k-optimal topologies[J]. arXiv preprint arXiv:2207.13604, 2022.
> >
> > Given the fact that the citations are indeed old, I hope the authors can look for more recent papers.
> >
> > 4. Example 1: Yes, I suggest remove it.
> >
> > 5. Position of related works section: After carefully re-read this section, I think does not explain the art of the related topics well. I'm aware that it is difficult to expand (as I mentioned above). So my suggestion is that the authors first revise it, and we discuss its position later.
> >
> > 6. The authors have understood my opinion. Sec 4.3 has been at the place for discussion. Therefore, if a proven statement is presented, then please put it into the conclusion section (to recall the contribution). If a conjecture is raised, then put it to a subsection entitled "discussion". The authors have clearly distinguished the role of each paragraph in the rebuttal.

---

> > > ### Author Response · Authors · 2023-08-18
> > >
> > > We really appreciate your suggestions how to improve the writing and organization of the manuscript. There are few things we do not fully understand in your response.
> > >
> > > 1. Can you be more specific in which aspect the supplementary did a poor job? We add the point about goal-awareness, since this is a difference between estimating cost-to-goal and ranking. We wanted to be honest that ranking losses as they does not make the heuristic goal-aware.
> > >
> > > 2. From your response, we did not figured out, how the reference [1] is related to the fact that ranking is easier than regression. This is cause that we not at all familiar with the non-homotopic path and it is going to take us while to understand it.  We know that our citations are old, nevertheless they very well carry the message. We wanted to find something recent, but were so far not successful. With that respect, we are grateful for your reference.
> > >
> > > 3. The related section is indeed brief, because the conference paper has a tight page limit. And although it would be really nice to write a longer text about the prior art, it would not fit the page limit, and honestly, it would be a different paper (something more like a survey). We therefore cited all works we were aware of that proposed something related to out work. We would be happy to know more details of how would you improve or what you do not like on the related work section.
> > >
> > > If we understand correctly, your criticism is related to the writing and structural aspect of the paper. The suggestions makes a lot of sense and they will help the paper, but they are relatively easy to incorporate and we would be more than happy to implement them for the final version of the paper.  We would also like to ask, if there is something regarding the technical and experimental contribution of the work that you do not like?
> > >
> > > Thanks a lot in advance for your response and time. We appreciate your comments, as they are generally good advices in scientific writing.

---

> > > > ### Comment · Reviewer_7xum · 2023-08-19
> > > >
> > > > Thank you for the response.
> > > >
> > > > 1. Related works issue: In my opinion, the proposed algorithm is trying to optimally rank the nodes in the priority queue of a searching-based algorithm. The precise estimation of heuristics of the nodes in the queue is a sufficient condition of obtaining the optimal ranking of the nodes in the queue (which is also a sufficient condition of the optimal choice of the next best node to be expanded). Hence, instead of estimating heuristics, estimating the ranking is an easier task. This is true. On the other hand, in [1], the algorithm compared the relative optimality of the searching branches in non-homotopic topological routes, which is very similar to the definition of "ranking". The elements that were compared in [1] are not X-Y locations of the robot but different homotopy classes of paths. I also think the authors might not be familiar with classic planning which is OK. But given the lack of related works I hope the authors re-do a careful searching of relative literatures.
> > > >
> > > > 2. Writing: I don't think the writing issues in this paper are easy to fix. If they are, then the authors should have fixed them before the submission. Given the fact that the authors did not even show me anything about how the manuscript is improved during this rebuttal phase, I have no further comments regarding this.

---

> > > > > ### Author Response · Authors · 2023-08-19
> > > > >
> > > > > Thanks for clarification of 1, we will give it a try.
> > > > >
> > > > > How would you like us to show that we are happy to incorporate your suggestions to the manuscript, when we cannot update the manuscript during the rebuttal phase?

---

### Official Review · Reviewer_jMYD · 2023-07-07

**Soundness:** 3 good
**Presentation:** 3 good
**Contribution:** 3 good
**Rating:** 7
**Confidence:** 4

**Summary:**

The paper considers the problem of using imitation learning for obtaining planning heuristics suitable for using variations of best-first search with a weighted sum of h() and g(), covering A* and GBFS. The paper gives both theoretical and empirical arguments for predicting a ranking of actions instead of predicting h*, the optimal distance to the goal. One argument in favour of ranking is that regression is a harder problem than regression. Another one is the observation that even with an optimal heuristic, the search can still expand more nodes than necessary. For that, in the case of symmetries, the proposed algorithm would choose a particular optimal solution among all the possible ones. The empirical results show the proposed approach to perform the best in A* in comparison with the others. A heuristic trained for GBFS performs well.

**Strengths:**

- Interesting discussion of ranking vs value estimation.
- Hard-to-achieve goal of expanding strictly nodes along the path.
- Discussion on the pros and cons of the approach.
	- Including an honest commentary on sample complexity.
- Promising empirical results.

**Weaknesses:**

- The proposed approach does not account for flexible goals or multiple goals.
- Limited discussion on the cases of symmetries and multiple optimal paths.
	- This is discussed but it could be analyzed further in the experiments.

**Questions:**

- What happens in the case of symmetry or multiple paths?
	- For instance, suppose there are k discrete integer variables, that can be increased and decreased independently. Suppose the goal is to set them to 0. Optimal plans just interleave increasing the variables independently. What would the theoretical analysis and the empirical results look like?
	- This example might be unrealistic but it's related to problems with very flexible solutions like the PDDL benchmark Logistics.
- How sensitive is the method to the distribution of problems and trajectories?
	- Is there any reason for it behaving differently across the proposed methods and the others?

**Limitations:**

The line of work is interesting, and there are certain domains where this method is relevant. The paper might benefit from exploring better those cases, given that the paper admits that there are cases where estimating the value might be better.

It'd be good to discuss the stability of training for the different algorithms.

The space used for introducing the search algorithm could be used for further discussion.

---

> ### Author Rebuttal · Authors · 2023-08-03
>
> Thank you for an interest in our work and great questions.
>
> *What happens if there are multiple solution path with the same length of the plan (number of actions) and cost?** The behavior of the resulting heuristic function depends on the composition of the training set. Let us illustrate the behavior on a simple grid problem. The agent starts at position (4,4) and has to move to the goal position (0,0). There are no obstacles and agent can only move one tile down or one tile right (the effects on his positions are either (-1,0) or (0,-1)), the cost of the action is one.  The number of different solutions path grows exponentially with the size of the grid and all solution path has the same cost (for problem of size 5x5, there are 70 solutions). This problem is interesting, since merit function in A* with optimal heuristic function (cost to goal) is constant, equal to 8, for all states.
>
> For the size of the grid (4,4), the heuristic is determined by table with 25 values. Below, we learn these values by minizing the proposed loss for A* algorithm with logistic loss surrogate. The minimization is carried by LBFGS (with all zeros as initial solution). Since the loss function is convex, LBFGS finds the solution quickly.
>
> In the first case, the training set contains one solution path ([4, 4], [3, 4], [2, 4], [1, 4], [0, 4], [0, 3], [0, 2], [0, 1], [0, 0]), where the agent first goes left and then down. The learnt heuristic values $h$ are
> ```
> ┌─────┬─────────┬─────────┬────────┬────────┬────────┐
> │ y\x │       0 │       1 │      2 │      3 │      4 │
> ├─────┼─────────┼─────────┼────────┼────────┼────────┤
> │  4  │ -191.53 │ -131.99 │ -76.94 │ -31.25 │    0.0 │
> │  3  │     0.0 │   31.25 │  76.94 │ 131.99 │ 191.53 │
> │  2  │     0.0 │     0.0 │    0.0 │    0.0 │    0.0 │
> │  1  │     0.0 │     0.0 │    0.0 │    0.0 │    0.0 │
> │  0  │     0.0 │     0.0 │    0.0 │    0.0 │    0.0 │
> └─────┴─────────┴─────────┴────────┴────────┴────────┘
>  ```
> An A* search using these heuristic values will first always take action moving the agent to left and then down. When moving down, he does not have other choice. Notice that heuristic values of many states are not affected by the optimization, because they are not needed to effectively solve the problem.
>
> In the second case, the training set will contain two solution path: the first is as above and the second the agent goes first down and then left. The learnt heuristic values are
> ```
> ┌─────┬────────┬───────┬────────┬────────┬────────┐
> │ y\x │      0 │     1 │      2 │      3 │      4 │
> ├─────┼────────┼───────┼────────┼────────┼────────┤
> │  4  │ -95.76 │ -66.0 │ -38.47 │  80.14 │    0.0 │
> │  3  │    0.0 │ 15.62 │  38.47 │ 131.99 │  80.14 │
> │  2  │    0.0 │   0.0 │    0.0 │  38.47 │ -38.47 │
> │  1  │    0.0 │   0.0 │    0.0 │  15.62 │  -66.0 │
> │  0  │    0.0 │   0.0 │    0.0 │    0.0 │ -95.76 │
> └─────┴────────┴───────┴────────┴────────┴────────┘
> ```
> An A* search will now face ties at state (4,4), since states (3,4) and (4,3) have the same heuristic value. But the presence of the tie does not affect the optimal efficiency of the search, as A* will always expand states on one of the optimal path from the training set.
>
> Finally let's consider a case, where all 70 possible solutions are in the training set. The learnt heuristic values are
> ```
> ┌─────┬─────────┬─────────┬─────────┬────────┬────────┐
> │     │       0 │       1 │       2 │      3 │      4 │
> ├─────┼─────────┼─────────┼─────────┼────────┼────────┤
> │  4  │    3.74 │   92.62 │  134.66 │ 163.17 │    0.0 │
> │  3  │  -46.61 │    2.35 │   91.93 │ 134.38 │ 163.17 │
> │  2  │ -167.96 │   -47.7 │    1.94 │  91.93 │ 134.66 │
> │  1  │ -210.03 │ -168.66 │   -47.7 │   2.35 │  92.62 │
> │  0  │     0.0 │ -210.03 │ -167.96 │ -46.61 │   3.74 │
> └─────┴─────────┴─────────┴─────────┴────────┴────────┘
> ```
> Rank of heuristic values "roughly" corresponds to cost to goal. The reason, why some states are preferred over the others despite their true cost-to-goal is the same is that they appear in more solution path. As above, the A* search with learn heuristic values is strictly optimally efficient.
>
> We emphasize that heuristic values are learnt such that the state closest cost to goal are expanded, which is a tie-breaking strategy for A* with exact cost to goal heuristic function.
>
> We hope that the above example clarifies the behavior of the proposed loss in cases, where the training set contains multiple optimal solutions path of the same problem instance. If the learnt heuristic learn ties, they do not affect the efficiency of the resulting search.
>
> *How sensitive is the method to the distribution of problems and trajectories?**The trajectories that are more frequent in the training set will be preferred. This is demonstrated on the above example, where states close to diagonal has smaller heuristic values. But this does not affect the efficiency of the search.
>
> Heuristic values will be good on problems similar to those in the training set, but might be poor on very different problems. But this problem is inherent to most flexible heuristic functions learn by machine learning methods. We emphasize that according to our experimental results, heuristic functions optimizing the proposed loss generalize better.
>
> *Is there any reason for it behaving differently across the proposed methods and the others?* We are not sure, what the reviewer meant by this question. The proposed loss is theoretically correct in the sense it optimizes the efficiency of finding an optimal solution. It is independent of the design of heuristic functions, which we have shown in the paper by learning heuristic functions for grid and relation (PDDL) domains. Furthermore, in the example in rebuttal, the heuristic function was realized by Table. So we do not have a reason to believe it will behave differently.

---

> > ### Comment · Reviewer_jMYD · 2023-08-20
> >
> > Thank you for your answers. The example is very insightful. After reading this and other responses, I'm increasing my rating to accept.
> >
> > Apologies for the bad writing of my question, 'Is there any reason for it behaving differently across the proposed methods and the others?'. I meant to add details to the question you answered 'How sensitive is the method to the distribution of problems and trajectories?', so it could have been phrased as "Is there any reason for the algorithm to behave differently across different datasets?".
> > I'm satisfied with this answer: 'The trajectories that are more frequent in the training set will be preferred. '

---

> > > ### Author Response · Authors · 2023-08-20
> > >
> > > Thank you very much. We appreciate it.

---

### Decision · Program_Chairs · 2023-09-21

**Decision:**

Accept (poster)

**Comment:**

The paper provides an interesting insight regarding the construction of planning heuristics, which, despite being largely theoretical and immediately applicable only to deterministic search problems, may be empirically useful in probabilistic planning or even RL as well. The reviewers and the AC trust that the authors will incorporate the changes that came out of their discussions with the reviewers into the final version.